# Correcting a bias in a climate model with an augmented emulator

Doug McNeall[1,3], Jonny Williams[2], Richard Betts[1,3], Ben Booth[1], Peter Challenor[3], Peter Good[1], and Andy Wiltshire[1,3]

[1]Met Office Hadley Centre, FitzRoy Road, Exeter, EX1 3PB, UK
[2]NIWA, 301 Evans Bay Parade, Hataitai, Wellington 6021, New Zealand
[3]University of Exeter, North Park Road, Exeter, EX4 4QE, UK

**Correspondence:** Doug McNeall (doug.mcneall@metoffice.gov.uk)

**Abstract.** A key challenge in developing flagship climate model configurations is the process of setting uncertain input parameters at values that lead to credible climate simulations. Setting these parameters traditionally relies heavily on insights from those involved in parameterisation of the underlying climate processes. Given the many degrees of freedom and computational expense involved in evaluating such a selection, this can be imperfect leaving open questions about whether any subsequent simulated biases result from mis-set parameters or wider structural model errors (such as missing or partially parameterised processes). Here we present a complementary approach to identifying plausible climate model parameters, with a method of bias correcting subcomponents of a climate model using a Gaussian process emulator that allows credible values of model input parameters to be found even in the presence of a significant model bias.

A previous study (McNeall et al., 2016) found that a climate model had to be run using land surface input parameter values from very different, almost non-overlapping parts of parameter space to satisfactorily simulate the Amazon and other forests respectively. As the forest fraction of modelled non-Amazon forests was broadly correct at the default parameter settings and the Amazon too low, that study suggested that the problem most likely lay in the model's treatment of non-plant processes in the Amazon region. This might be due to modelling errors such as missing deep-rooting in the Amazon in the land surface component of the climate model, to a warm-dry bias in the Amazon climate of the model, or a combination of both.

In this study we bias correct the climate of the Amazon in the climate model from (McNeall et al., 2016) using an "augmented" Gaussian process emulator, where temperature and precipitation, variables usually regarded as model outputs, are treated as model inputs alongside land surface input parameters. A sensitivity analysis finds that the forest fraction is nearly as sensitive to climate variables as it is to changes in its land surface parameter values. Bias correcting the climate in the Amazon region using the emulator corrects the forest fraction to tolerable levels in the Amazon at many candidates for land surface input parameter values, including the default ones, and increases the valid input space shared with the other forests. We need not invoke a structural model error in the land surface model, beyond having too dry and hot a climate in the Amazon region.

The augmented emulator allows bias correction of an ensemble of climate model runs and reduces the risk of choosing poor parameter values because of an error in a subcomponent of the model. We discuss the potential of the augmented emulator to act as a translational layer between model subcomponents, simplifying the process of model tuning when there are compensating errors, and helping model developers discover and prioritise model errors to target.

## 1 Introduction

### 1.1 Choosing good input parameter settings in the presence of model errors

Choosing values of uncertain input parameters that lead to credible climate simulations is an important and challenging part of developing a new climate model configuration. Climate models contain simplifications of processes too complex to represent explicitly in the model, termed parameterisations. Associated with these parameterisations are coefficients called input parameters, the values of which are uncertain and can be set by the model developer. We wish to choose input parameters where the output of the model reproduces observations of the climate, in order to have confidence that the model represents important physical processes sufficiently well to trust projections of the future. This is difficult because: 1) There is uncertainty in the observations, 2) we cannot run the model at every desired input parameter configuration, and there is uncertainty about model output at those parameter sets not run, and 3) the model does not reproduce the dynamics of the climate system perfectly. The latter is termed model discrepancy, and distinguishing between it and a poorly chosen input parameter configuration is a major challenge in model development.

Input parameters have a material effect on the way the parameterisations operate and therefore induce an uncertainty in the output of the model and corresponding uncertainty in projections of future climate states, but often to an extent that is unknown until the model is run. Modern climate simulations are computationally expensive to run, and there may only be a handful of simulations on which to make a judgement about the validity of a simulation at a particular set of parameters. Further, appropriate values for input parameters may be difficult or even impossible to observe, some having no direct analogue in the real system.

Setting input parameters traditionally relies heavily on insights from those involved in parameterisation of the underlying climate processes. Given the many degrees of freedom and computational expense involved in evaluating such a selection, this can be an imperfect process, leaving open questions about whether any subsequent simulated biases result from mis-set parameters or wider structural model errors (such as missing or partially parameterised processes). The process of setting the values of the input parameters so that the simulator output best matches the real system is called tuning, and where a probability distribution is assigned for the input parameters, it is termed calibration. This process is often viewed as setting constraints on the plausible range of the input parameters, where the climate model sufficiently represents the real system.

In summarising current practice in the somewhat sparsely studied field of climate model tuning, Hourdin et al. (2017) point out that it remains an art as well as a science. While there appear no universally accepted procedures, individual modelling centres have begun to document their tuning practices (Schmidt et al., 2017; Zhao et al., 2018; Walters et al., 2017).

Improving a coupled climate model can require an involved and lengthy process of development, and parameter tuning occurs at different stages in that process. It might start with a single column version of the model developed in isolation as stand-alone code. It can be relatively easy to find a good subset of input parameters given a small set of inputs and outputs and a well behaved relationship between the two as for a subcomponent of a climate model, particularly where there are good

observations of the system being studied. The climate model components to be coupled might then be tuned with standard boundary conditions - for example tuning a land/atmosphere component with fixed or historically observed sea surface temperatures. Finally, a system-wide tuning might be used to check that there are minimal problems once everything has been coupled together. There is sometimes a tension however, between choosing input parameters that elicit the best performance for the subcomponent (e.g. for a single-gridbox model), and choosing ones that make the subcomponent behave well in the context of the coupled model. Upon integration, some components of the model may therefore be tuned to compensate for errors in others or there may be unknown errors in the model or observations. Golaz et al. (2013) show the potential impact of compensating errors in tuning. They find that two different but plausible parameter configurations of the cloud formations of the coupled climate model GFDL-CM3 can result in similar present-day radiation balance. The configurations did not differ in their present day climate, but showed significantly different responses to historical forcing and therefore historical climate trajectories. More complex models are computationally expensive and so are infeasible to run in enough configurations to be able to identify these kind of errors. No single expert, or even small team of experts may have the cross-domain knowledge required to identify and fix problems that occur as multiple sub systems interact with each other. Output from a climate model run at a particular set of inputs must be evaluated against observational targets of the real system. Individual observations are subject to uncertainties, sometimes large, and there are often multiple observations of the same property, each with its own strengths and weaknesses.

Without information about known errors (for example, knowledge of an instrument bias, or a known deficiency of a model), it can be difficult to attribute a difference between simulator output and the real system to underlying model errors, to an incorrect set of input parameters, or to inaccuracies in the observations. This means that good candidates for input parameters might be found in a large volume of input space, but projections of the model made with candidates from across that space might diverge to display a very wide range of outcomes. This problem is sometimes referred to as "identifiability", but otherwise known as "equifinality", or the "degeneracy" of model error and parameter uncertainty.

Although climate model tuning is overall a subjective process, individual parts of the process are amenable to more algorithmic approaches. Statistical and machine learning approaches to choosing parameters to minimise modelling error, or to calculate probability distributions for parameters and model output are known as uncertainty quantification (UQ). The field of Uncertainty Quantification (UQ) has seen a rapid development of methods to quantify uncertainties when using complex computer models to simulate real, physical systems. The problem of accounting for model discrepancy when using data to learn about input parameters is becoming more widely recognised in UQ. It was formalised in a Bayesian setting by Kennedy and O'Hagan (2001). The authors suggested simultaneously estimating a model discrepancy - there called model inadequacy - as a function of the inputs, using a Gaussian process prior. Brynjarsdóttir and O'Hagan (2014) argued that only by accounting for model discrepancy does even a very simple simulator have a chance of making accurate predictions. Further, they found that only where there is strong prior evidence about the nature of that model discrepancy is it possible to solve the inverse problem and recover the correct inputs. Without this strong prior evidence the estimate of the correct parameters is likely to be overconfident, and wrong, leading to overconfident and wrong predictions of out-of-sample data. Arendt et al. (2012a) offer a

number of examples of identifiability problems, ranging from solvable using mild assumptions through to virtually impossible. In a companion paper (Arendt et al., 2012b), they outline a way of improving identifiability using multiple model responses.

## 1.2 History matching

Some of the dangers of overconfident and wrong estimates of input parameters and model discrepancy can be reduced using a technique called history matching (Craig et al., 1996), sometimes called pre-calibration or iterated refocussing. The aim of history matching is not to find the most likely inputs, but to reject those unlikely to produce simulations statistically close to observations of the real system.

A statistical model called an emulator, trained on an ensemble of runs of the climate model, predicts the output at input configurations not yet run. An implausibility measure ($I$) is calculated at any input configuration, taking into account the distance between the simulator output and the observation but formally allowing for uncertainty in the observations, the simulator output and the simulator discrepancy. Those inputs that produce a large implausibility score are ruled out from consideration as candidate points. New simulator runs in the remaining input space increase our understanding of the model behaviour and allow more input space to be ruled out in an iterated fashion. An excellent introduction and case studies can be found in Andrianakis et al. (2015), or in Vernon et al. (2010). History matching is perhaps less ambitious but correspondingly more robust than calibration methods, and a full calibration can be carried out once the history matching procedure has been completed.

History matching can be effective in reducing the volume of parameter space that is considered plausible to produce model runs that match the real system. For example, Williamson et al. (2015) report very large reductions (around 99%) in the volume of space considered plausible. History matching does still depend however, on a robust estimate of model discrepancy and associated uncertainty, in order not to produce unjustifiably small regions of not-ruled-out input parameter space. For example, McNeall et al. (2013) studied an ensemble of an ice sheet model and found that using a single type of observation for ruling out input space was not very powerful - particularly if there was not a very strong relationship between an input parameter and the simulator output. The effectiveness of history matching for ruling out input space can be enhanced by using multiple data sets. However Johnson et al. (2018), using history matching to constrain the forcing of a coupled climate and atmospheric chemistry model, find that even with multiple observational targets, a typical example of aerosol effective radiative forcing is only constrained by about 30%.

McNeall et al. (2016) argued that the larger the number of model-data comparisons, the larger the probability that an unidentified model discrepancy rules out a perfectly good input parameter candidate point. Several empirical rules have been used in the literature - for example using the maximum implausibility of a multiple comparison, a candidate input point may be ruled out by a single observation. A more conservative approach is to use the second or third implausibility score, or to use a multivariate implausibility score, both introduced in Vernon et al. (2010). The aim of these scores is to ensure that an unidentified model discrepancy, or a poorly specified statistical model of the relationships between model inputs and outputs does not result in ruling out candidate points that are in fact perfectly good.

While history matching has often been used to explore and reduce the input parameter space of expensive simulators, its use as a tool to find discrepancies, bias and inadequacies in simulators is less developed. Williamson et al. (2015) argue that what

was assumed a structural bias in the ocean component of climate model HadCM3 could be corrected by choosing different parameters. In a different system McNeall et al. (2016) argue that a standard set of parameters for the land surface component of the climate model FAMOUS should be retained, and that a bias seen in the simulation of the Amazon rainforest is a simulator discrepancy not a poor parameter choice. When cast as a choice between adding a model discrepancy and keeping the default parameters, or rejecting them and accepting the new region of parameter space, they argued that the former was more likely to produce a good model for a number of reasons, whereas were a number of reasons one might reject the proposed parameter space. First, scientific judgement, expertise and experience with previous versions of this and other models will have informed the original choice of parameters. The model simulated other forests at the standard set of parameters well, and only a tiny volume of parameter space could be found that (barely) adequately simulated all the forests. The region of parameter space that apparently simulated all forests well was at the edge of the ensemble, where uncertainty in the emulator is often large, and might dominate the history matching calculation rather than the parameter choices being particularly good. In that case, running more ensemble members in the part of parameter space in question might help rule it out. Three of the forests were well simulated at the default parameters and a highly overlapping region of parameter space, and only the Amazon was poorly simulated at the default parameter setting. Finally, in the region where all forests were adequately simulated, the Amazon forest was under estimated, and the other forests overestimated, suggesting that that choosing that region of parameter space would inevitably force a compromise.

## 1.3 Aims of the paper

This paper revisits and extends the analysis of McNeall et al. (2016) (hereafter M16) to attempt to identify the source of model discrepancy in the simulation of the Amazon in FAMOUS. We aim to 1) identify the causes of a low bias in the forest fraction in the Amazon region in an ensemble of the climate model FAMOUS and 2) develop a method that allows us to choose plausible values for input parameters for one component of a coupled model, even when there is a model discrepancy or bias in another subcomponent of the coupled model.

A well simulated and vigorous Amazon forest at the end of the spinup phase of a simulation experiment is a prerequisite for using a model to make robust projections of future changes in the forest. The analysis of M16 identified that the land surface input spaces where FAMOUS forest fraction was consistent with observations were very different in the Amazon than they were for other forests. The area of overlap of these spaces - one that would normally be chosen in a history matching exercise - did not simulate any of the forests well, and did not contain the default parameters. M16 suggested that assuming an error in the simulation of the Amazon forest would be a parsimonious choice. Two obvious candidates for the source of this error in the Amazon region were identified: (1) a lack of deep rooting in the Amazon forest, meaning that trees could not access water at depth as in the real forest and (2) a bias in the climate of the model, affecting the vigour of the trees.

We simultaneously (1) assess the impact of a bias corrected climate on the Amazon forest and (2) identify regions of input parameter space that should be classified as plausible, given a corrected Amazon climate. To bias correct the climate we develop a new method to augment a Gaussian process emulator, with simulator outputs acting as inputs to the emulator alongside the standard input parameters. We use simulated output of forests at different geographical locations to train the

emulator, describing a single relationship between the climate of the simulator, the land surface inputs and the forest fraction. In doing so, we develop a technique that might be used to bias correct subcomponents of coupled models, allowing a more computationally efficient method for final system-tuning of those models.

In section 2, we review the literature on the possible causes of the low Amazon forest fraction in FAMOUS. In section 3.1, we describe how we use the temperature and precipitation to augment the Gaussian process emulator. In section 4.2 we use the augmented emulator to bias correct the climates of the forest and examine the effect of that bias correction on the input space that is deemed statistically acceptable in a history matching exercise. In section 4.5 we search for regions of parameter space where the bias corrected simulator might perform better than at the default parameters. In section 4.4 we use the augmented emulator to estimate the sensitivity of forest fraction to changes in land surface and climate parameters In section 4.6 we look at regions of climate space where the default parameters would produce statistically acceptable forests. Finally, we offer some discussion of our results in section 5 and conclusions in section 6.

## 2  Climate and forest fraction

Previous studies have concluded that the climate state has an influence on the Amazon rainforest. Much of that work has been motivated by the apparent risk of dieback of the Amazon forest posed by a changing climate [e.g. Malhi et al. (2008); Cox et al. (2004)]. We assume that factors that might affect a future simulated Amazon rainforest might also affect the simulated steady-state preindustrial forest in FAMOUS. Parameter perturbations and $CO_2$ concentrations have been shown to influence the simulation of tropical forests in climate models (Boulton et al., 2017; Huntingford et al., 2008), with increases in $CO_2$ fertilisation and associated increased water use efficiency through stomatal closure offsetting the negative impacts of purely climatic changes (Betts et al., 2007; Good et al., 2011). A metric linked to rainforest sustainability by Malhi et al. (2009) is Maximum Cumulative Water Deficit, which describes the most negative value of climatological water deficit measured over a year. In a similar vein Good et al. (2011, 2013) find that in Hadley Centre models, sustainable forest is linked to dry-season length, a metric which encompasses both precipitation and temperature, along with sensitivity to increasing $CO_2$ levels. No forest is found in regions that are too warm or too dry, and there is a fairly distinct boundary between a sustainable and non-sustainable forest. Galbraith et al. (2010) found that temperature, precipitation and humidity had greatly varying influences, and by different mechanisms on changes in vegetation carbon in the Amazon across a number of models, but that rising $CO_2$ mitigated losses in biomass. Poulter et al. (2010) found that the response of the Amazon forest to climate change in the land surface model LPJml was sensitive to perturbations in parameters affecting ecosystem processes, the carbon cycle and vegetation dynamics, but that the dynamics of a dieback in the rainforest was robust across those perturbations. In that case, the main source of uncertainty of dieback was uncertainty in climate scenario. Boulton et al. (2017) found that temperature threshold and leaf area index parameters both have an impact on the forest sustainability under projections of climate change in the Earth system version of HadCM3.

## 2.1 Biases in FAMOUS

M16 speculated that both local climate biases and missing or incorrect processes in the land surface model - such as missing deep rooting in the Amazon - might be the cause of the simulated low forest fraction in the Amazon region at the end of the pre-industrial period in an ensemble of the climate model FAMOUS. In this study we use the ensemble of FAMOUS previously used in M16, to attempt to find and correct the cause of persistent low forest fraction in the amazon, identified in that paper.

The Fast Met Office UK Universities Simulator, FAMOUS (Jones et al., 2005; Smith et al., 2008), is a reduced-resolution climate simulator based on the climate model HadCM3 (Gordon et al., 2000; Pope et al., 2000). The model has many features of modern climate simulators, but is of sufficiently low resolution to provide fast and simple data sets with which to develop UQ methods. Full details of the ensemble can be found in M16 and Williams et al. (2013).

The ensemble of 100 members perturbed 7 land surface and vegetation inputs (see supplementary material, table S1), along with a further parameter denoted "beta" ($\beta$). Each of the ten values of beta provides a index to one of ten of the best-performing atmospheric and oceanic parameter sets used in a previous ensemble with the same model Gregoire et al. (2010), with the lowest values of beta corresponding to the very best performing variants. The beta parameter therefore summarised perturbations in 10 atmospheric and oceanic parameters that impacted the climate of the model, randomly varied with land surface input parameters, and potentially leading to different climatologies in a model variant with the same land surface parameters but different values of beta. Variations in the beta parameter did however not correlate strongly to variations with any of the oceanic, atmospheric or land surface parameters in the ensemble, and so the parameter was excluded from the analysis in M16. In this analysis we recognise that the different model climates caused by variations in the atmospheric and oceanic parameters will have an impact on the forest fraction, and so we summarise those variations directly using local temperature and precipitation.

Variation in the parameters across the ensemble had a strong impact on vegetation cover at the end of a spinup period, with atmospheric $CO_2$ at preindustrial conditions. The broadleaf forest fraction in individual ensemble members varies from almost non-existent to vigourous (fig. 1). The strong relationships between the global mean forest fraction and the mean forest fraction in each region implies that perturbations in input parameters exert a larger control over all forests simultaneously, and individual forests to a smaller extent.

M16 aggregated the regional mean forest fraction for the Amazon, Southeast Asian, North American and central African forests, along with the global mean. They were only able to find very few land surface parameter settings which the history matching process suggested should lead to an adequate simulations of the Amazon forests and the other forests together. These parameter sets were at the edges of sampled parameter space, where larger uncertainty in the emulator may have been driving the acceptance of the parameter sets.

In this study, we use the same ensemble of forest fraction data used in M16. However, we add temperature and precipitation data, present in the original ensemble but not used to build an emulator in the M16 study, to further our understanding of the causes of the low forest fraction in the Amazon region. The temperature and precipitation data summarise the effects of atmospheric parameters on the atmospheric component of the model, in a way that is directly seen by the land surface

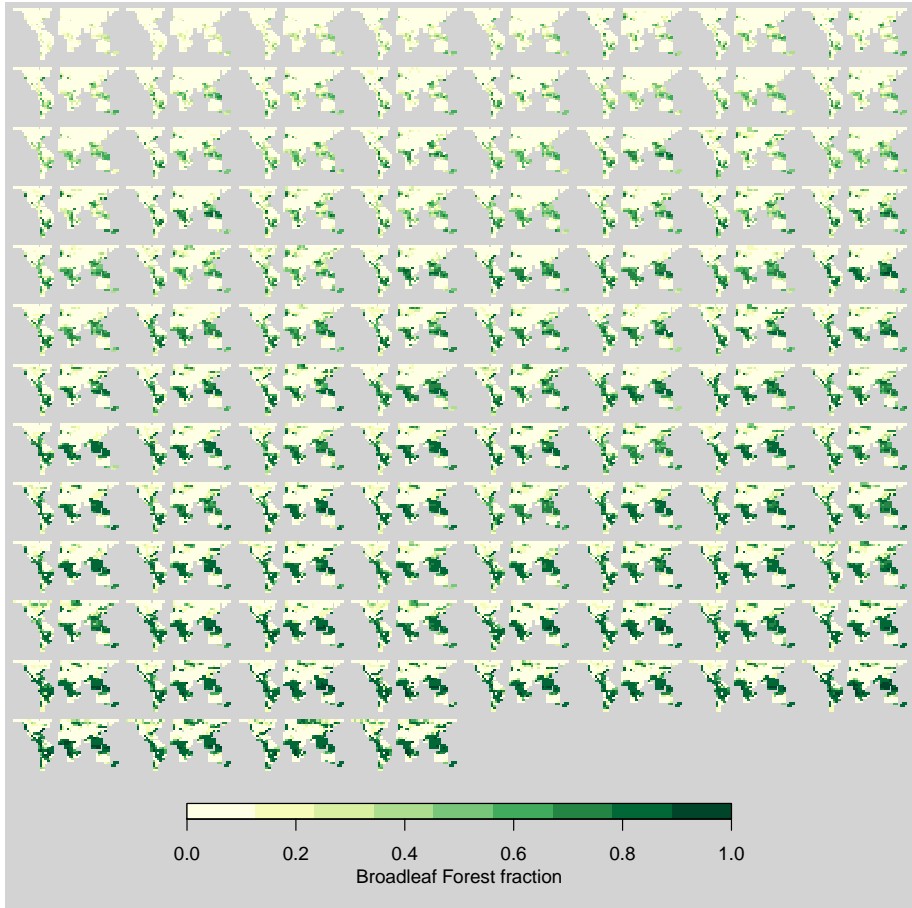

**Figure 1.** Broadleaf forest fraction in the FAMOUS ensemble, ranked from the smallest to largest global mean value.

component of the model. We consider only regions dominated by tropical broadleaf forest, so as not to confound analysis by including other forests which may have a different set of responses to perturbations in parameters, rainfall and temperature.

For temperature observations we use the CRU global monthly surface temperature climatology (Jones et al., 1999), covering the years 1960-1990. For precipitation we use the average monthly rate of precipitation, covering the years from 1979-
5   2001 from GPCP Version 2.2 provided by the NOAA/OAR/ESRL PSD, Boulder, Colorado, USA, from their Web site at https://www.esrl.noaa.gov/psd/ (Adler et al., 2003). Vegetation fraction observations are adapted from Loveland et al. (2000), and are shown in fig. 2. Although the observations all cover slightly different time periods, we expect the differences caused by harmonising the time periods to be very small compared to other uncertainties in our analysis, and to be well covered by our uncertainty estimates.

10   A plot of regional mean temperature and precipitation in the tropical forest regions in the FAMOUS ensemble (fig. 3) indicates the form of the impact that the regional climate has on forest fraction in the climate model. Central African and

Southeast Asian climates in the model simulations run in a sweep across the middle of the plot, from dry and cool to wet and warm.

It appears that a wetter climate - which would be expected to stabilise forests - broadly compensates for the forest reductions induced by a warmer climate. Within the ensemble of Central African forests for example, forest fraction increases towards the "cooler, wetter" (top left) part of the climate phase space. Beyond a certain value however, there are no simulated climates or forests in this climatic region. It is clear from the plot that while central African and Southeast Asian forests are simulated in the large part considerably warmer than recent observations, they are also simulated considerably wetter, which might be expected to compensate forest stability. In contrast, while simulated considerably warmer, the Amazon is also slightly drier than recent observations, which might further reduce forest stability.

We are assuming here that tropical forests can be represented by a single set of forest function parameters. While such an assumption risks missing important differences across heterogenous tropical forests, modelling the system with the smallest set of common parameters avoids overfitting to present day data. Avoiding overfitting is important if we are to use these models to project forest functioning in future climates outside observed conditions. One of the questions that the analysis presented in this paper addresses is whether current forest biases in the simulations reflect limitations of this single tropical forest assumption, or whether biases in the simulations of the wider climate variables play a more important role.

## 3    Methods

The climate model FAMOUS is computationally expensive enough that we cannot run it for a large enough number of input parameter combinations to adequately explore parameter space and find model biases. To increase computational efficiency we build a Gaussian process emulator: a statistical function that predicts the output of the model at any input, with a corresponding estimate of uncertainty (see e.g. Sacks et al. (1989); Kennedy and O'Hagan (2001)). The emulator models climate model output $y$ as a function $g()$ of inputs $x$ so that $y = g(x)$. It is trained on the ensemble of model runs described in section 2.1. The set of land surface input parameters is called the design matrix, denoted $X$, and the corresponding sample of model output forest fraction is denoted $y$. The configuration of the design matrix is a Latin hypercube (McKay et al., 1979), as used in e.g. Gregoire et al. (2010); Williams et al. (2013), with sample input points chosen to fill input parameter space efficiently and therefore sample relationships between input parameters effectively.

### 3.1    An augmented emulator

Our strategy is to augment the design matrix of input parameters $X$ with corresponding atmospheric climate model output that might have an impact on the modelled land surface, building an emulator that models the effects of both input parameters and climate on forest fraction. We then use the augmented emulator to bias correct each forest in turn. We use the emulator to describe the relationship between land surface parameters, atmospheric variables that summarise the action of hidden atmospheric parameters, and the broadleaf forest fraction. The relationships between these variables are summarised in fig. 4.

We have a number of forests for each ensemble member, differing in driving influences by a different local climate. Regional extent of each of the broadleaf forests can be found in the supplementary material. We use regional mean temperature, $T$, and precipitation, $P$, for each of the forests: the Amazon, central Africa and Southeast Asia as additional inputs to augment our original design matrix of land surface parameters, $X$. These new inputs are outputs of the model when run at the original

inputs $X$, and are influenced by the 10 atmospheric and oceanic parameters perturbed in a previous ensemble in a configuration unavailable to us in this experiment. Performance of the model under those perturbations is summarised in the beta parameter, which has smaller values for the better performing models. The performance metrics included temperature and precipitation, along with a number of other measures so the beta parameter therefore contains information about temperature and precipitation across the ensemble, without being a perfect representation of its behaviour. We cannot control the atmospheric and oceanic

parameters directly and thus ensure that they lie in a Latin hypercube configuration, although the ensemble is ordered in a Latin hypercube configuration according to the performance of the model at each parameter set.

With $n = 100$ ensemble members, we form each $n \times 1$ vector of temperature and precipitation and form an $n \times 2$ matrix of climate variables for the Amazon $C_{AZ} = [T_{AZ} P_{AZ}]$, Central Africa $C_{AF} = [T_{AF} P_{AF}]$ and Southeast Asia $C_{AS} = [T_{AS} P_{AS}]$. We use these to augment the original $n \times p$ input matrix $X$, creating a unique input location for each forest. We then stack these

augmented input matrices together to form a single input matrix $X'$.

$$X' = \begin{bmatrix} X & C_{AZ} \\ X & C_{AF} \\ X & C_{AS} \end{bmatrix} \tag{1}$$

From an initial ensemble design matrix with $n = 100$ members and $p = 7$ inputs, we now have a design with $n = 300$ members and $p = 9$ inputs. Each member with a replicated set of initial input parameters (e.g members $[1, 101, 201]$), differ only in the $T$ and $P$ values. Figure 5 shows a diagram of the augmented emulator along with the composition of the resulting

input matrix and output vector.

Where in M16, the authors built an independent emulator for each output (i.e. regional forest fraction), we now build a single emulator for all forest fractions simultaneously given input parameters, temperature and precipitation. The output vector for the tropical forests has gone from being 3 sets of 100 values $y_{AZ}, y_{AF}, y_{AS}$, to a single vector $y' = [y_{AZ}, y_{AF}, y_{AS}]$ of length 300. We model forest fraction $y'$ as a function of $X'$ using the Gaussian process emulator of package DiceKriging (Roustant

et al., 2012) in the R statistical language and environment for statistical computing. Details of the emulator can be found in the supplementary material.

We note that the augmented emulator depends on the assumption that modelled broadleaf forests in each location respond similarly to perturbations in climate and input parameters. This assumption may not hold for the behaviour of the forests in the model, or indeed the real world. For example, particularly deep rooting of forests in the Amazon would respond differently to

rainfall reductions but these processes are not represented in the underlying climate model. Similarly, differing local topology that is captured in the climate model, may influence the forests in a way not captured by our emulator. In both cases, the emulator would show systematic errors of prediction.

**Table 1.** Mean absolute error (MAE) rounded to the first significant figure for the regular emulator, using just the seven land surface inputs, and the augmented emulator, including temperature and precipitation.

| Forest | Regular emulator MAE | Augmented emulator MAE |
|---|---|---|
| Amazon | 0.05 | 0.03 |
| Southeast Asia | 0.06 | 0.03 |
| Central Africa | 0.06 | 0.03 |
| All | 0.06 | 0.03 |

## 3.2  Verifying the augmented emulator

To verify that the augmented emulator adequately reproduces the simulator behaviour, we use a leave-one-out metric. For this metric, we sequentially remove one simulator run from the ensemble, train the emulator on the remaining ensemble members and predict the held-out run. We present the predicted members and the calculated uncertainty plotted against the actual ensemble values in fig. 6.

It is important to check that the augmented emulator performs well in prediction, in order to have confidence that using emulated runs in our later analyses is a valid strategy. We see no reason to doubt that the augmented emulator provides a good prediction and accurate uncertainty estimates for prediction at inputs points not yet run. We use the mean of the absolute value of the difference between the emulator prediction and corresponding held-out value to calculate the Mean Absolute Error of cross-validation prediction (MAE). Prediction error and uncertainty estimates remain approximately stationary across all tropical forests and values of forest fraction. The mean absolute error of prediction using this emulator is a little under 0.03, or around 6% of the mean value of the ensemble.

When compared against the regular emulator using just the land surface inputs, the augmented emulator performs well. The regular emulator built individually for each of the forests (as per M16) has a mean absolute error value of 0.058 - nearly double that of the augmented emulator. This indicates that adding temperature and precipitation to the input matrix adds useful information to a predictive statistical model. A breakdown of the mean absolute error of the emulator on a per-forest basis can be seen in table 1.

There is some concern that the emulator might not perform well close to the observed values of temperature and precipitation, particularly for the Amazon and Central African regions. For this reason, we carry out an enhanced verification of the emulator, holding out more ensemble members and demanding further extraoplation (see supplemantary material, section 2). We find no reason to doubt that the augmented emulator performs well.

Do the error estimates of the augmented emulator match the true error distributions when tested in leave-one-out predictions? We test the reliability of uncertainty estimates of the emulator by checking that the estimated probability distributions for held-out ensemble members match the true error distributions in the leave-one-out exercise. We create a rank histogram (see e.g. Hamill (2001)) for predictions, sampling 1000 times from each Gaussian prediction distribution, and plotting the

rank of the actual prediction in that distribution. The distribution of these ranks overall predictions should be uniform if the uncertainty estimates are reliable. Consistent overestimation of uncertainty will produce a peaked histogram, while systematic underestimation of uncertainty will produce a u-shaped histogram. The rank histogram produced by this set of predictions (fig. 7) is close to a uniform distribution, indicating reliable predictions.

## 3.3 History Matching

History matching is the process of finding and ruling out regions of parameter space where the model is unlikely to produce output that matches observations well. It measures the statistical distance between an observation of a real-world process, and the emulated output of the climate model at any input setting. An input where the output is deemed too far from the observation is ruled "implausible", and removed from consideration. Remaining inputs are conditionally accepted as "Not Ruled Out Yet" (NROY), recognising that further information about the model or observations might yet rule them as implausible.

Observations of the system are denoted $z$, and we assume that they are made with uncorrelated and independent errors $\epsilon$ such that

$$z = y + \epsilon \tag{2}$$

Assuming a "best" set of inputs $x^*$ where the model discrepancy $\delta$, or difference between climate model output $y$ and $z$ is minimised, we relate observations to inputs with

$$z = g(x^*) + \delta + \epsilon \tag{3}$$

We calculate measure of implausibility $I$, and reject any input as implausible where $I > 3$ after Pukelsheim's three-sigma rule; that is, for any unimodal distribution, 95 % of the probability mass will be contained within 3 standard deviations of the mean (Pukelsheim, 1994). We calculate

$$I^2 = |z - E[g(x)]|^2 / Var[g(x)] + Var[\delta] + Var[\epsilon] \tag{4}$$

which recognises that the distance between the best estimate of the emulator and the observations must be normalised by uncertainty in the emulator $g(x)$, in the observational error $\epsilon$, and in the estimate of model discrepancy $\delta$.

## 4 Analyses

### 4.1 The joint impacts of temperature and precipitation on forest fraction

What impact do temperature and precipitation have on forest fraction together? We use the emulator from section 3.1 and predict the simulator output across the entire range of simulated temperature and precipitation, while holding the other inputs

at their default values. The marginal impacts of temperature and precipitation on forest fraction are clear in fig. 8. Ensemble member temperature, precipitation and forest fraction, taken from fig. 3 are overplotted for comparison. Temperature and precipitation values are normalised to the range of the ensemble in this plot.

With other inputs held constant, cooler, wetter climates are predicted to increase forest fraction and drier, warmer climates reduce forest fraction. In general, South East Asian and Central African forests are simulated as warmer and wetter than their true-life counterparts. Moving the temperature and precipitation values of a typical ensemble member from near the centre of these forest sub-ensembles to their observed real-world values would shift them primarily in the same direction as the contours of forest fraction value. This would mean that bias correcting the climate variables would not have a large impact on forest fraction values in South East Asian and Central African forests, and that they are therefore simulated with a roughly accurate forest fraction. In contrast, the Amazon is simulated slightly drier, and considerably warmer than the observed Amazon and many ensemble members consequently have a lower forest fraction than observed. Shifting the temperature and precipitation of a typical ensemble member for the Amazon to its real-world observed values would cross a number of countours of forest fraction. This figure provides strong evidence that a significant fraction of the bias in Amazon forest fraction is caused by a bias in simulated climate.

## 4.2  A climate bias correction approach

With an emulator that models the relationship between input parameters, local climate and the forest fraction, we can predict what would happen to forest fraction in any model simulation if the local climate was correct. In fig. 9, we compare the value of forest fraction predicted at the default set of land surface parameters using the standard emulator, with that predicted using the local temperature and precipitation corrected to the observed values using the augmented emulator. This means that Central Africa becomes significantly drier, and a little cooler than the centroid of the ensemble. Southeast Asia becomes a little cooler and a little drier. The Amazon forest becomes a little wetter, and significantly cooler. The ensemble has a much larger spread of climates in central Africa than South East Asia or the Amazon. We note that we do not have an ensemble member run at the default land surface parameters, so we compare two predictions using the emulator.

The bias correction reduces the difference between the prediction for the modelled and observed Amazon forest fraction markedly, from -0.28 using the standard emulator to -0.08 using the augmented emulator. It makes the predicted modelled forest in central Africa worse (-0.11 from -0.03), and slightly improves the SE Asian forest fraction (0.07 from 0.1). Overall, bias correcting the climate takes the mean absolute error at the default parameters from 0.14 to 0.09 for the three forests. It is possible that the predicted forest fraction for central Africa is slightly worse because the observed climate is towards the edge of the parameter space of temperature and precipitation, and there are no runs near.

## 4.3  History matching to learn about model discrepancy

In this section we use history matching (see section 3.3) to learn about parts of input parameter space that are consistent with observations, and to find the causes of discrepancy in the model. We study the region of input space that is "Not Ruled Out Yet" (NROY) by comparison of the model output to the observations of forest fraction. In the previous section we see that the

**Table 2.** Mean absolute error of the simulated forest fraction, and Implausibility of the default set of land surface parameters when non-bias corrected and bias corrected to temperature and precipitation observations.

| Forest | Error | Implausibility | Error (bias corrected) | Implausibility (bias corrected) |
|---|---|---|---|---|
| Amazon | 0.316 | 6.94 | -0.079 | 1.31 |
| Southeast Asia | -0.096 | 1 .61 | 0.072 | 1.76 |
| Central Africa | -0.04 | 0.768 | -0.11 | 1.5 |

overall difference between the simulated and observed forest fraction is reduced if the output is bias corrected. In this section, we study how that bias correction affects the NROY space.

In M16, the default input parameters were ruled out as implausible for the Amazon region forest fraction. For the sake of illustration, we assume very low uncertainties: zero observational uncertainty and a model discrepancy term with a zero mean and an uncertainty ($\pm$ 1 sd) of just 0.01. We note that under these conditions the default parameters would be ruled out in the standard emulator. However, if we bias correct the model output using the observed temperature and precipitation, we find that the implausibility measure $I$ for the forest fraction in the Amazon at the standard input parameters reduces from nearly 7 to 1.3 - comfortably under the often-used threshold of 3 for rejection of an input. The implausibility of the SE Asian and Central African forest fraction at the default parameter settings rises with bias correction (see table 2), but neither comparison comes close to ruling out the default parameters. This rise in implausibility is caused by a smaller uncertainty estimate (in the case of SE Asia), and a larger emulation error (in the case of Central Africa). However, we can confidently say that bias correction using the emulator means that observations no longer rule out the default parameters, even with the assumption of a very small model discrepancy.

Another result of bias correction is that it increases the "harmonisation" of the input spaces - that is, the volume of the input space that is "shared", or Not Ruled Out Yet by any of the comparisons of the simulated forest fractions with data. In M16, we argued that the regions of input parameter space where the model output best matched the observations had a large shared volume for the Central African, Southeast Asian and North American forests. In contrast, the "best" input parameters for the Amazon showed very little overlap with these other forests. This pointed to a systematic difference between the Amazon and the other forests that might be a climate bias, or a fundamental discrepancy in the land surface component of the model. Here, we show that the climate-bias-corrected forest in the Amazon would share a much larger proportion of its NROY space with the other forests. Indeed, the default parameters are now part of this "shared" space, and there is formally no need to invoke an unexplained model discrepancy in order to accept them for all the tropical forests. We show a cartoon of the situation in fig. 10.

We find that when we bias correct all the spaces, the proportion of "shared" NROY input space relative to the union of NROY spaces for all forests increases from 2.6% to 31% - an order of magnitude increase (see table 3). This is driven chiefly by the harmonisation of the NROY space of the Amazon to the other two forests. We see that before bias correction, the South East Asian and African forests share nearly three quarters (74%) of their combined NROY space. This drops to 33% when bias

**Table 3.** Measures of the NROY input space shared by all three forests. The intersection is NROY for all three forests, the Union is NROY for at least one forest. The initial space is that defined by the parameter limits of the initial experiment design.

|  | Intersection / Union (%) | Intersection / Initial (%) |
|---|---|---|
| Non bias-corrected | 2.6 | 1.9 |
| Bias-corrected | 31 | 28.3 |

corrected, but with the advantage that the Amazon and Central Africa now share over 90% (91.5%) of their combined NROY space (table 4).

When compared to the initial input parameter space covered by the ensemble, the shared NROY space of the non-bias-corrected forests represents 1.9%, rising to 28% on bias correction.

We visualise two dimensional projections of the NROY input parameter space shared by all three forests before bias correction in fig. 11 and after bias correction in 12. The two dimensional projections of high density regions of NROY points are dramatically shifted and expanded in the bias corrected input space, and the default parameters now lie in a high density region. For example, a high-density region of NROY points is apparent in the bias-corrected input parameter space (fig. 12), in the projection of V_CRIT_ALPHA and NL0. It is clear from sensitivity analyses (sect. 4.4) that, all other things remain-

ing the same, increasing the value of NL0 raises strongly forest fraction, while increasing V_CRIT_ALPHA strongly reduces forest fraction. We would expect there to be a region, indeed a plane through parameter space where these two strong effects counteract each other, resulting in a forest fraction close to observations. This feature does not appear in the history matching before bias correcting (fig. 11). The low value of the simulated Amazon forest fraction before bias correction of the climate inputs rules out much of the input parameter space later found to be Not Ruled Out Yet (NROY) after the bias corrected history

matching exercise (fig. 12).

It is possible that the estimate of shared NROY input space is larger than it could be, due to the lack of ensemble runs in the "cool, wet" part of parameter space, where there are no tropical forests. Inputs sampled from this part of parameter space may not be ruled out, as the uncertainty on the emulator may be large. This is history matching working as it should, as we have not included evidence about what the climate model would do if run in this region. Further work could explore the merits

of including information from other sources (for example, from our knowledge that tropical forests do not exist in a cool wet climate) into the history matching process.

## 4.4 Sensitivity analysis

The augmented emulator allows us to measure the sensitivity of forest fraction to the land surface input parameters simultaneously with climate variables temperature and precipitation. A quantitative measure of sensitivity of the model output to

25 parameters that does take into account interactions with other parameters is found using the FAST99 algorithm of Saltelli et al. (1999), summarised in fig. 13. Precipitation and Temperature are the second and third most important parameters, more impor-

**Table 4.** Proportion of shared NROY input space for each forest pair compared to the total NROY space covered by the same forest pair. Non bias corrected (top) and bias corrected (bottom).

| Non bias-corrected | Amazon | Southeast Asia | Central Africa |
|---|---|---|---|
| Amazon | 1 | | |
| Southeast Asia | 0.034 | 1 | |
| Central Africa | 0.075 | 0.741 | 1 |
| Bias-corrected | | | |
| Amazon | 1 | | |
| Southeast Asia | 0.329 | 1 | |
| Central Africa | 0.915 | 0.337 | 1 |

tant than NL0, and only slightly less important than V_CRIT_ALPHA. Interaction terms contribute a small but non-negligible part to the sensitivity. This form of quantitative sensitivity analysis is useful to understand initial model behaviour, but could be vulnerable to error, as it is assumed that all parts of the input space are valid. Our experiment design does not control temperature and precipitation directly, and the "cool, wet" part of this parameter space does not contain tropical broadleaf forest. It is

5 possible therefore that a sensitvity analysis that relies on input samples from this region might mis-specify sensitivity indices. Below, we outline two methods that tackle this problem: One-at-a-time sensitivity analysis with history matching, and Monte Carlo filtering.

We measure the one-at-a-time sensitivity to parameters and climate variables, using the augmented emulator to predict changes in forest fraction as each input is changed from the lowest to highest setting in turn, with all other inputs at the default

settings or observed values. We present the results in fig. 14. In this diagram, we exclude emulated forests that are deemed implausible according to the criteria in section 3.3. This is to avoid potentially over-estimating the sensitivity of forest fraction to (for example) temperature and precipitation by including results from regions of parameter space far from existing ensemble members, and in very different climate regimes from existing broadleaf tropical forests.

Parameters NL0 and V_CRIT_ALPHA and climate variables temperature and precipitation exert strong influences of similar

magnitudes on forest fraction. Shaded regions represent the uncertainty of the sensitivity to each parameter, due to estimated emulator uncertainty of $\pm$ 2 standard deviations. This sensitivity measure does not include the extra uncertainty due to the fact that the relationships will change depending on the position of the other parameters. We do however see a measure of how temperature and precipitation affect the marginal response of the other parameters, as the observed climates of each forest are different. For example, we clearly see that the response of the forest fraction to e.g. NL0 depends on climate - the forest

fraction response is a noticeably different shape when varied under the mean climate of the South East Asian region.

A technique called Monte Carlo Filtering (MCF), or Regional Sensitivity Analysis is useful in situations where input parameter distributions are non-uniform, correlated, or not all parts of parameter space are valid. The basic idea of MCF is to

split samples from the input space into those where the corresponding model output meets (or not) some criteria of behaviour. Examining the differences between the cumulative distributions of those inputs where the outputs do or do not meet the criteria provides a measure of sensitivity of the output to that input. For example, we might split model behaviour into those outputs above or below a threshold. A recent description of MCF and references can be found in section 3.4 of Pianosi et al. (2016).

We integrate the MCF sensitivity analysis into the history matching framework. We examine the differences in the univariate cumulative distributions of each parameter, in those samples where the output is ruled out by history matching, against those that are "Not Ruled Out Yet" (NROY). To measure the differences between the distributions we perform a two-sided Kolmogorov–Smirnov (KS) test and use the KS statistic as in indicator that the output is sensitive to that input. A larger KS statistic indicates that the cumulative distribution function of the respective inputs are further apart, that that input is more im-

portant for determining if the output falls within the NROY part of parameter space, and therefore the output is more sensitive to that input in a critical region. We note that MCF is useful for ranking parameters, but not for screening, as inputs that are important only in interactions might have the same NROY and ruled out marginal distributions. In this case they would have a sensitivity index of zero.

    We apply MCF using the emulator. This allows us to examine the difference between model output distributions given a

15 much larger sample from the input space than when using only the ensemble. This comes at the cost of using an imperfect emulator, which may give different results than if we were using a large ensemble of runs. To avoid the problem of sampling precipitation and temperature from regions where there are no ensemble members, we sample uniformly from across input space for all other parameters, and then append a random temperature/precipitation location from the ensemble. We calculate a sampling uncertainty by calculating the MCF sensitivity metrics 1000 times, each time using a sample size of 5000 emulated

ensemble members. In this way, we estimate both the mean and the uncertainty (standard deviation) of the MCF sensitivity measures. We note that the sensitivity indices are calculated higher when a small number of ensemble members are used, as well as with a higher uncertainty. The change in both the estimated statistic and its uncertainty have begun to become small by the time 3000 ensemble members are used, suggesting that we should use at least this many emulated ensemble members to obtain an unbiased sensitivity analysis (see supplementary material). We compare the KS statistics and their associated

uncertainty for each input in fig. 15.

    We can check the strength of the relationship between the MCF sensitivity measures and the FAST99 sensitivity measures, by plotting them together. We examine this relationship in the supplementary material (fig. S7).

## 4.5    Doing better than the default parameters

We can use the emulator to find locations in parameter space where there is a potential that the difference between the modelled

and observed forest fractions could be smaller than at the default parameters. Figure 16 shows the density of parameter settings in each 2-dimensional projection of the input space, where the emulator estimates the model performs better than at the default parameters, once bias correction has been applied. That is, the absolute difference between each estimated forest fraction and the observed values is smaller than the absolute difference of the mean estimate at the default parameters. Out of 100000 samples from the uniform hypercube defined by the range of the experiment design, only 2451, or around 2.5%

match this criterion and are plotted. This diagram might help guide further runs in the ensemble, choosing high density regions to run new ensemble members. The convergence of NL0 and V_CRIT_ALPHA seems particularly focussed, and suggests that a lower value of V_CRIT_ALPHA might be a way to reduce error in the forest fraction. There is another, although less densely populated region of high NL0 and V_CRIT_ALPHA that might fulfil the criteria of lower estimates of error for each forest. These regions would be good targets for supplementary runs of the climate model, and for particularly careful emulator checking. A poorly performing emulator could guide a model developer into wasting model runs at locations which, in reality, did not produce forest fractions close to the observed values.

## 4.6 Allowable climate at default parameters

We use history matching to find the set of regional mean climates that are most consistent with the observations for each tropical forest. To illustrate the best case scenario we set model discrepancy, its associated uncertainty, and observational uncertainty artificially low (0, 0.01 and 0 respectively), so that implausibility is almost exclusively a product of the emulator uncertainty. We find the set of NROY temperature and precipitation values when the remaining input parameters are held at their default values. Figure 17 shows the density of NROY points in the climate space for each of the observed forest fractions. We see that the Amazon and Central African forests might be well simulated in the model in a very wide range of cooler and wetter climates, with only the "hot, dry" corner showing zero density of potential inputs that produce similar forest fraction to observations. The Southeast Asian forest fraction is matched by a swathe of inputs running diagonally through the centre of input space. Neither the hot-dry or cool-wet corners of input space produce forests that match the observations, though the warm-wet and cool-dry corners do.

## 5 Discussion

### 5.1 Simulating the Amazon

We have shown that the simulation of the broadleaf tropical forest in FAMOUS is almost as sensitive to temperature and precipitation as to any land surface parameter perturbation in the ensemble. However, the calculated sensitivities are dependent on the chosen limits of the parameter perturbations themselves. The precise order and size of sensitivities might change given updated parameter ranges, but there is little doubt that the climate variables are a strong influence on broadleaf forest fraction. This version of FAMOUS when run with the default land surface input parameter settings would successfully simulate the Amazon rainforest to within tolerable limits if regional climate biases were substantially reduced. As such, there is no need to invoke a missing process in the land surface in order to explain the forest fraction discrepancy in the Amazon. We have strengthened the case made by M16 that the low Amazon forest fraction is not a result of poorly chosen parameters. There is a broad region of climate space where the effects of temperature and precipitation on forest fraction compensate for each other. This gives room for a number of possible sources of model discrepancy, and by extension makes it unlikely that the default

input parameters are optimal. There are indications from the emulator that a small region of parameter space exists where there is even smaller overall error in the simulation, offering a target for exploration using further runs of the model.

There is a feedback from the land surface to the atmosphere implicitly included in the emulated relationship. We cannot control this feedback directly with the emulator, and so work out the impact of this feedback on the forest fraction as it is present in the training data. This feedback would have to be taken into account if we were to simulate the correct climate independently of the land surface.

It is possible that were we to include a process seen to be missing from the Amazon (such as deeper rooting of trees allowing them to thrive in drier climates), our map of NROY input space would alter again. Given that there is a measure of uncertainty in observations and the emulator, as well as the possibility of further compensating errors, we cannot rule out a model discrepancy such as a deep rooting process. The fact that the other forests do slightly less well when their climates are bias corrected points to a potential missing process in the model, compensated for by parameter perturbations. However, the impact of this missing process is likely much smaller than we might have estimated had we not taken the bias correction of the forest into account.

## 5.2   Uses for an augmented emulator

By building an emulator that includes temperature and precipitation - traditionally used as climate model outputs - we are able to separate the tuning of one component of the model (here the atmosphere) from another (the land surface). Perturbations to the atmospheric parameters, tested in a previous ensemble but not available to us except through an indicator parameter, are summarised as inputs through the climate of the model.

We have used the augmented emulator as a translational layer between components of the model. The augmented emulator allows us to ask "what would it mean for our choice of input parameters if the mean climate of the model in the Amazon region were correct?" This means that we will have less chance of ruling out parts of parameter space that would lead to good simulations or keeping those parts that lead to implausible simulations. An augmented emulator as a translational layer might be built as part of a model development process, making it computationally cheaper and faster. Traditionally, the components of computationally expensive flagship climate models are built and tuned in isolation before being coupled together. The act of coupling model components can reveal model discrepancies or inadequacies. A model discrepancy in one model component can mean that a connected subcomponent requires retuning from its independently-tuned state. There is a danger that this retuning leads to a model that reproduces historical data fairly well, but that makes errors in fundamental processes and therefore is less able to predict or extrapolate - for example, a climate model when projecting future changes under unprecedented greenhouse gas concentrations. Given the time and resources needed to run such complex models, these errors might persist much longer than necessary, and have profound consequences for climate policy.

A translational layer would allow parameter choices to be made for a model when run in coupled mode, even when there was a significant bias in one of the components that would affect the other components. The translational layer would bias correct the output of a component of the model, allowing an exploration of the effects of input parameter changes on the subcomponent of the model, in the absence of significant errors. Using the augmented emulator could eliminate some of the steps in the tuning

process, help the model developer identify potential sources of bias, and to quickly and cheaply calculate the impacts of fixing them. In doing so it would aid model developers in identifying priorities for and allocating effort in future model development.

Our work here shows this process as an example. We have identified the importance of precipitation and temperature to the correct simulation of the Amazon forest, and flag their accurate simulation in that region as a priority in for the development of any climate model that hopes to simulate the forest well. We have identified regions of the space of these climate variables where the Amazon forest might thrive, and related that back to regions of land surface parameter space that might be targeted in future runs of the model. We have achieved this in a previously-run ensemble of the model, allowing computational resources to be directed towards new climate model runs that will provide more and better information about the model.

There are also potential computational efficiencies in our approach of decoupling the tuning of two components (here the atmosphere and the land surface) in the model. A good rule of thumb is that a design matrix for building an emulator should have $O(10 \times p)$ training points, where $p$ is the number of input parameters, in order to adequately sample parameter space to the extent it is possible to build a good emulator. With approximately 10 atmospheric and 7 land surface parameters, we would need $O(170)$ runs. Here, we have summarised those 10 parameters as two outputs that have a material impact on the aspect of the land surface that we are interested in. Adding these two to the 7 inputs, we need $O(10 \times (2 + 7) = 90)$ runs, well covered by our available ensemble of 100 runs.

We acknowledge however that in order to trace back information about the performance of the model in forest fraction to the original 10 oceanic and atmospheric parameters, we would need access to the original ensemble. We have used temperature and precipitation to reduce the dimension of the parameter space, but there is no guarantee that the relationship between the original parameters and the local climate is unique. There may be multiple combinations of the 10 parameters that lead to the temperature and precipitation values seen, which would mean that we would require a large ensemble to estimate the relationships well. Alternatively, there may be an even more efficient dimension reduction for forest fraction, meaning we would need even fewer model runs to summarise the relationship.

## 5.3 Limitations

In theory the augmented emulator could be used to bias correct differently sized regions, down to the size of an individual gridbox for a particular variable. This might be useful for correcting, for example, known biases in elevation or seasonal climate. The principle of repeating the common parameter settings in the design matrix, and including model outputs as inputs would work in exactly the same way, but with a larger number of repeated rows. In the case of using an augmented emulator on a per-gridbox basis, we might expect the relationship between inputs that we are bias correcting (e.g. temperature, precipitation), and the output of interest (e.g. forest fraction) to be a less clear, as at small scales there are potentially many other inputs that might influence the output. An emulator for an individual gridbox might therefore be less accurate. However, with enough data points, or examples (and there would be many), we might expect to be able to recover any important relationships.

The computational resources needed to fit a Gaussian process emulator when the number of outputs estimated simultaneously becomes even moderately large limits the use of our technique. The design input parameter matrix used for training the emulator grows to $n \times d$ rows, where $n$ is the number of ensemble members in the original training set, and $d$ is the number of separate

output instances to be considered. In our example, $d$ is 3, and so we only have $100 \times 3 = 300$ in the new training set. Given an initial ensemble of a few hundred, this could easily result in a training set with hundreds of thousands or even millions of rows. Gaussian process emulators are currently limited to using training data with perhaps a few hundred rows as current software packages must invert an $n \times n$ matrix, a potentially very computationally expensive process (see e.g. Hensman et al. (2013) for examples). At the time of writing this limitation would preclude using our specific technique for correcting biases on a per-gridbox basis. To make use of the translational layer for large data sets we would need new Gaussian process technology, or specific strategies to deal with large data sets. These strategies might involve kernel based methods, keeping the scope of training data local to limit the size of any inverted matrices. Alternatively, they might involve building emulators using only a strategically sampled selection of the outputs. Recent advances in using Gaussian processes for larger data sets can be found in Hensman et al. (2013, 2015); Wilson et al. (2015); Wilson and Nickisch (2015). Our current strategy is to reduce the dimension of the output of the climate model, by taking the regional mean of the output of the climate model (temperature and precipitation). More advanced dimension reduction techniques might offer great potential.

Given that we overcome such technical barriers, we see no reason that such a layer not be built that is used to (for example) correct the climate seen by individual land surface grid boxes, rather than (as here) individual aggregated forests. The process of rejecting poor parameter sets might be aided by having a comparison against each gridbox in an entire global observed surface, rather than aggregated forests. Alternatively, we might allow parameters to vary on a gridbox-by-gridbox basis, effectively forming a map of Not-Ruled-Out-Yet parameters.

If trained on an ensemble of model runs which included all major uncertainties important for future forests, an augmented emulator could be used directly to estimate the impacts and related uncertainty of climate change on forest fraction in the model, even in the presence of a significant bias in a model subcomponent. After estimating the relationship between the uncertain parameters, climate, and the forest fraction, we could calculate the forest fraction at any climate, including those that might be found in the future. This ensemble of climate model runs would project the future forests under a number of atmospheric $CO_2$ concentrations and parameter combinations. It would be necessary that the training data included any climates that might be seen under the climate change scenario to be studied, as the emulator has much larger uncertainties if asked to extrapolate beyond the limits of the training data. The trajectory of vegetation states through time would also be an important element of the ensemble, as the vegetation state is path dependent. However there would be great potential to save a large number of runs, as not every parameter perturbation would have to be run with every projection scenario. Such a set of runs would serve as a framework upon which a great many post hoc analyses could be done with the emulator. Once the set of runs was complete, they would effectively serve as the definitive version of the model - any new information that needed to be extracted from the model could in theory be found using the emulator. Not only might we be able to identify and correct important climate biases and their impact on the forest, but also update our estimates of forest change as we learn more about the uncertainty ranges of the uncertain parameters and forcing trajectory.

## 6 Conclusions

A previous study (McNeall et al., 2016) concluded that it was difficult to simulate the Amazon rainforest and other tropical rainforests at a set of input parameters in the climate model FAMOUS, pointing to a climate bias or model discrepancy as a source of error. Here we demonstrate that we can correct the simulation of the Amazon rainforest in the climate model FAMOUS by correcting the regional bias in the climate of the model with a Gaussian process emulator. We therefore find it unnecessary to invoke a model discrepancy or inadequacy, such as a lack of deep rooting in the Amazon in the model, to explain the anomalously low forest fraction in an ensemble of forests simulations.

We present a method of augmenting a Gaussian process emulator by using climate model outputs as inputs to the emulator. We use average regional temperature and precipitation as inputs, alongside a number of land surface parameters, to predict average forest fraction in the tropical forests of the Amazon, Southeast Asia and central Africa. We assume that the differences in these parameters account for the regional differences between the forests, and use data from all three tropical forest regions to build a single emulator. We find that the augmented emulator improves accuracy in a leave-one-out test of prediction, reducing the mean absolute error of prediction by almost half, from nearly 6% of forest fraction to just under 3%. This allays any fears that the emulator is inadequate to perform a useful analysis, or produces a measurable bias in predictions, once augmented with temperature and precipitation as inputs. In two types of sensitivity analyses, temperature and precipitation are important inputs, ranking 2 and 3 after V_CRIT_ALPHA (rank 1) and ahead of NL0 (rank 4).

We use the augmented emulator to bias correct the climate of the climate model to modern observations. Once bias corrected, the simulated forest fraction in the Amazon is much closer to the observed value in the real world. The other forests also change slightly, with central Africa moving further from the observations, and Southeast Asia moving slightly closer. We find that the differences in the accuracy of simulation of the Amazon forest fraction and the other forests can be explained by the error in climate in the Amazon. There is no requirement to invoke a land surface model discrepancy in order to explain the difference between the Amazon and the other forests. After bias correction, the default parameters are classified as "Not Ruled Out Yet" in a history matching exercise, that is they are conditionally accepted as being able to produce simulations of all three forests that are statistically sufficiently close to the values observed in the real world. Bias correction "harmonises" the proportion of joint NROY space that is shared by the three forests. This proportion rises from 2.6% to 31% on bias correction. Taken together these findings strengthen the conclusion of McNeall et al. (2016) that the default parameters should not be ruled out as implausible by the failure of FAMOUS to simulate the Amazon. We find a small proportion (around 2.5%) of input parameter space where we estimate that the climate model might simulate the forests better than at the default parameters. This space would be a good target for further runs of the simulator.

We offer a technique of using an emulator augmented with input variables that are traditionally used as outputs, to aide the tuning of a coupled model perturbed parameter ensemble by separating the tuning of the individual components. This has the potential to (1) reduce the computational expense by reducing the number of model runs needed during the model tuning and development process and (2) help model developers prioritise areas of the model that would most benefit from development.

The technique could also be applied to efficiently estimate the impacts of climate change on the land surface, even where there are substantial biases in the current climate of the model.

*Code and data availability.* Code and data are available at https://doi.org/10.5281/zenodo.3246103

*Author contributions.* DM designed the analysis and wrote the paper with the assistance of all other authors. JW ran the climate model and provided the climate model data.

*Competing interests.* The authors declare no competing interests.

*Acknowledgements.* This work was supported by the Met Office Hadley Centre Climate Programme funded by BEIS and Defra. DM and AW were also supported by the Newton Fund through the Met Office Science for Service Partnership Brazil (CSSP Brazil). DM would like to acknowledge the Isaac Newton Institute programme workshop on Uncertainty Quantification for Complex Systems and its participants for useful discussions while writing this paper. We would like to thank David Sexton for insightful comments on the manuscript.

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

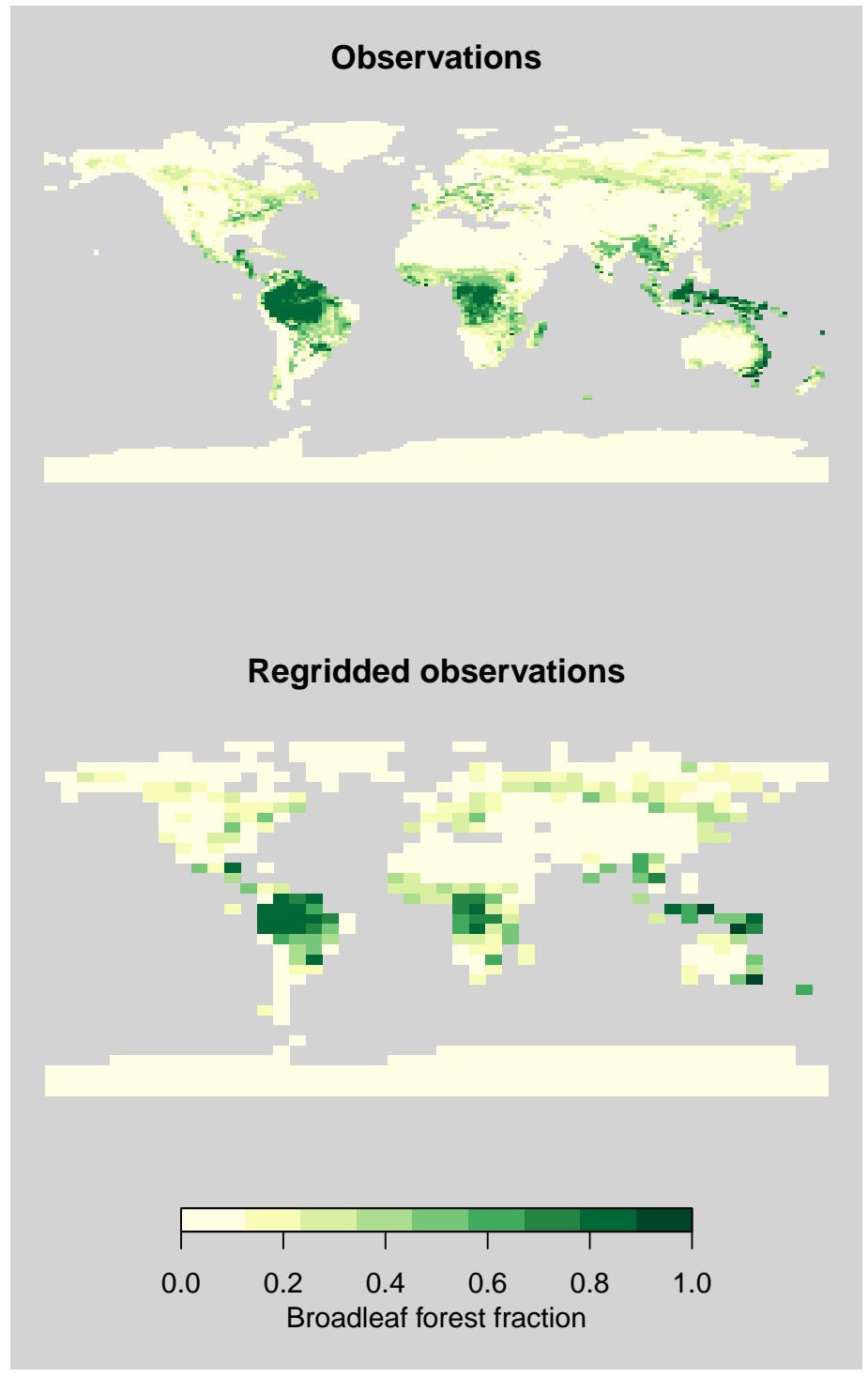

**Figure 2.** Observations of broadleaf forest fraction on their native grid (top), and regridded to the FAMOUS grid (bottom).

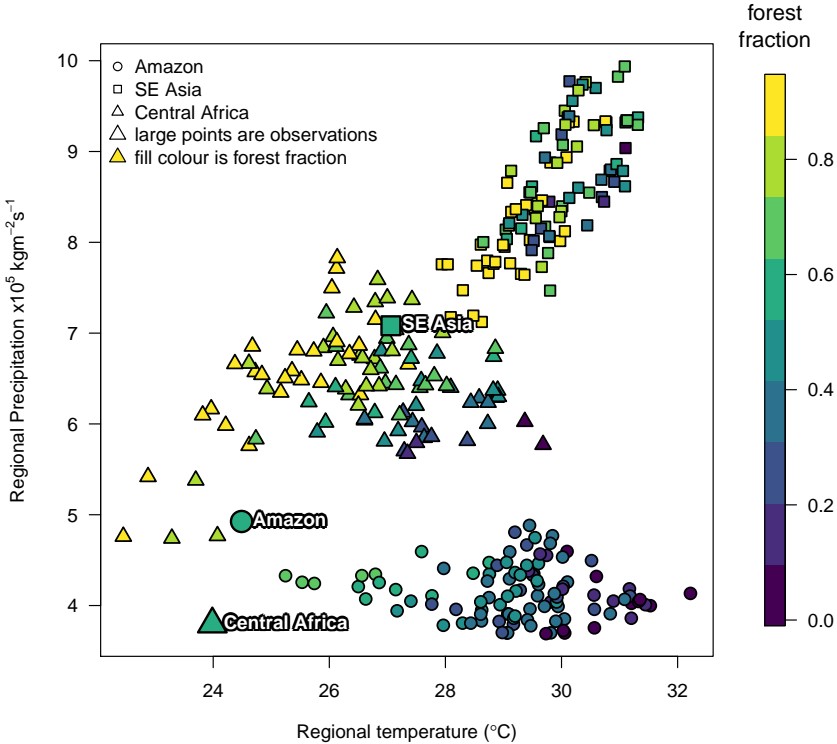

**Figure 3.** Regional temperature, precipitation and broadleaf forest fraction in the ensemble of FAMOUS compared with observations. Smaller symbols represent broadleaf forest fraction in the FAMOUS ensemble against regional mean temperature and precipitation. Ensemble member forest fraction in the Amazon is represented by the colour of the circles, Central Africa by triangles and SE Asia by squares. Larger symbols represent observed climate and forest fraction.

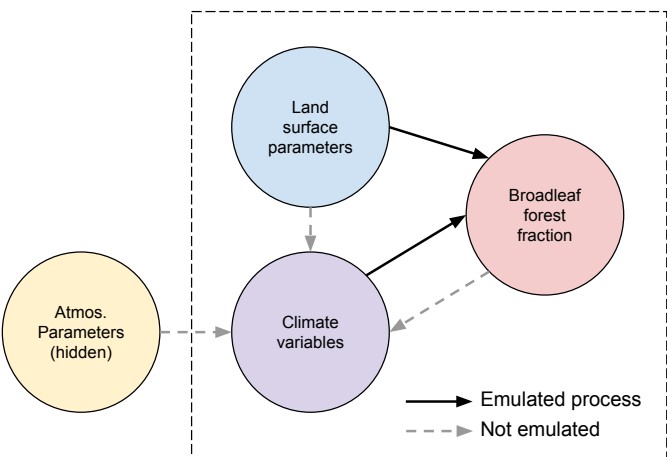

**Figure 4.** A graph showing the assumed relationship between input parameters, climate variables and forest fraction. An arrow indicates influence in the direction of the arrow. Processes that are directly emulated are shown with a solid arrow, while the processes shown by a dotted arrow are not directly emulated.

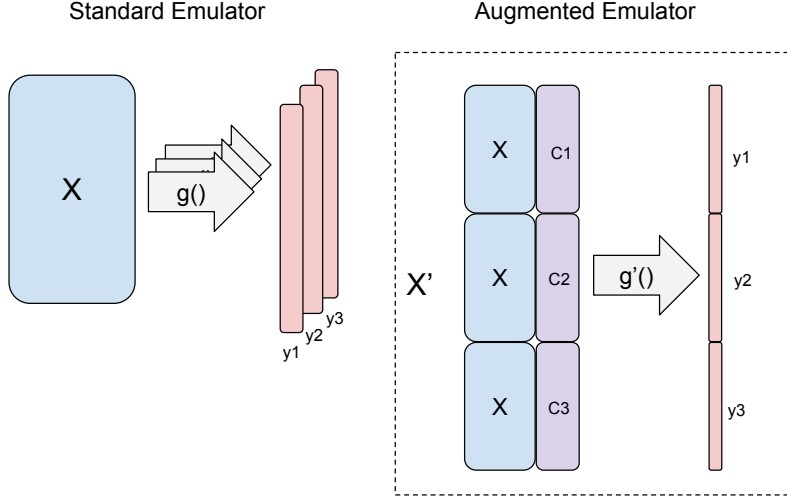

**Figure 5.** In a standard emulator setup (left), training data consists of an input matrix $X$ and corresponding simulator output $y$. A new emulator $g_1(), \ldots, g_3()$ is trained for each output $y_1, \ldots, y_3$ of interest. In the augmented emulator, output from the simulator $C_1, \ldots, C_3$ augments the design matrix, with the initial inputs $X$ repeated.

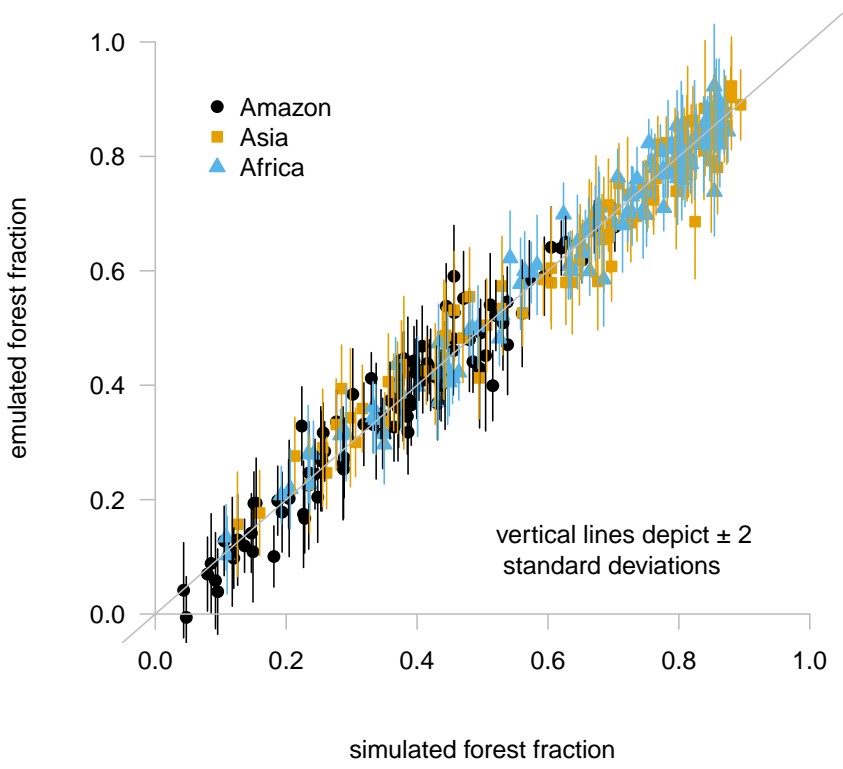

**Figure 6.** Leave-one-out cross validation plot, with the true value of the simulator output on the x-axis, and predicted output on the y-axis. Vertical lines indicate $\pm 2$ standard deviations.

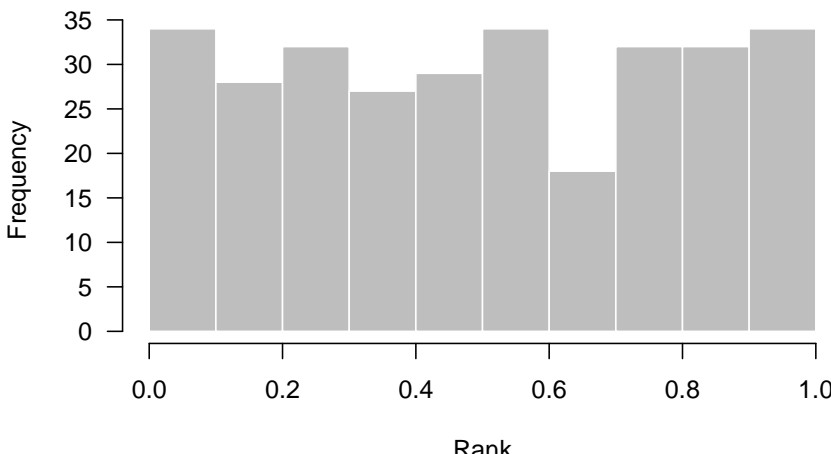

**Figure 7.** Rank histogram of leave-one-out predictions. For each prediction of a held-out ensemble member, we sample 1000 points from the Gaussian prediction distribution, and then record where the true held-out ensemble member ranks in that distribution. We plot a histogram of the ranks for all 300 ensemble members. A uniform distribution of ranks indicates that uncertainty estimates of the emulator are well calibrated.

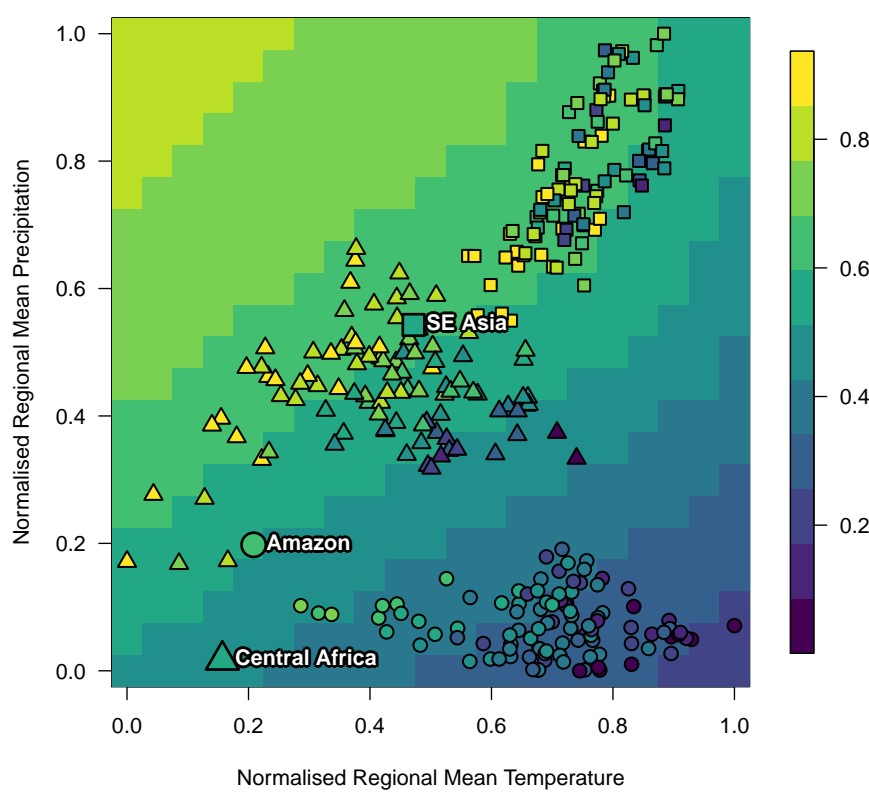

**Figure 8.** The impact of climate on forest fraction. Background plot colour indicates the mean emulated forest fraction when all land surface inputs are held at their default values. Temperature and precipitation in the ensemble are marked with symbols, with the fill colour representing forest fraction. Larger symbols represent the values observed in the real world.

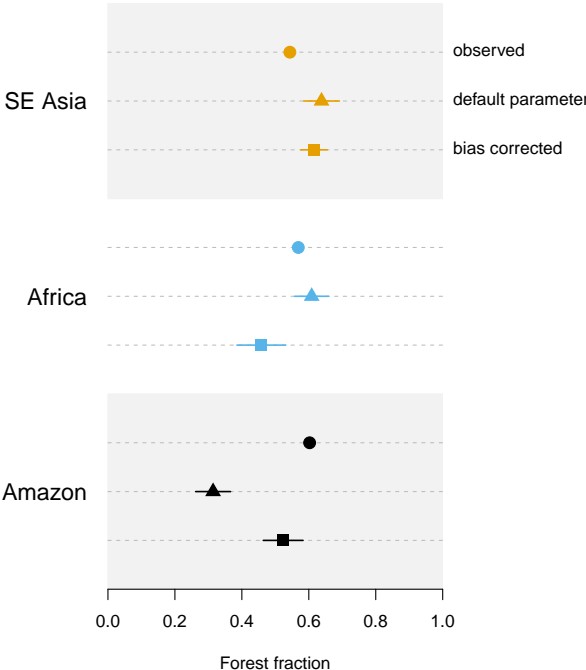

**Figure 9.** Observed and emulated Forest fraction in each tropical forest. For the emulated forest fraction at default and bias corrected parameters, emulator uncertainty of $\pm$ 2sd is represented by horizontal bars.

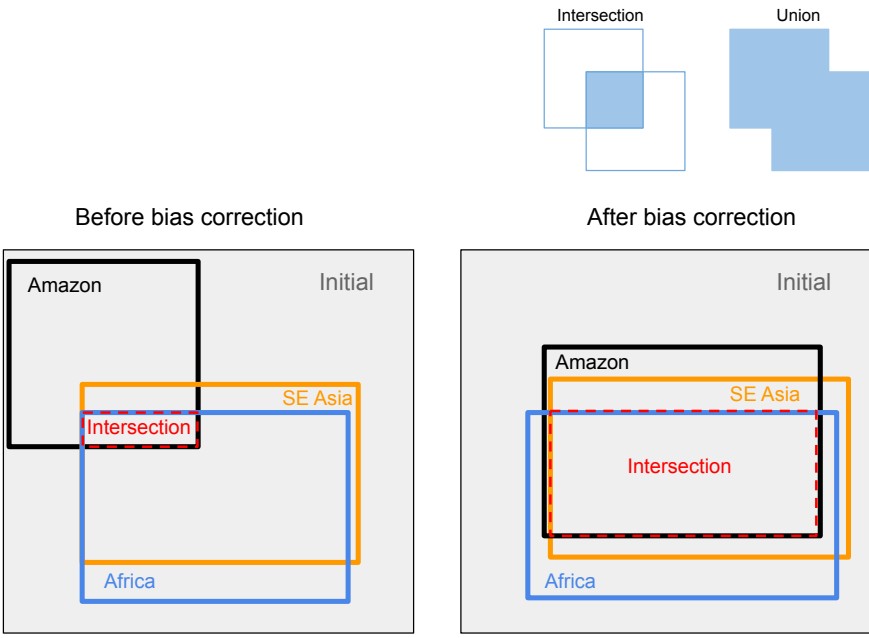

**Figure 10.** A cartoon depicting the input space that is "not ruled out yet (NROY)" when the climate simulator output is compared to observations of the forest fraction in the Amazon, Africa, and South East Asia before (left) and after (right) bias correction. We measure the "shared" space (the intersection of NROY spaces for each forest) as a fraction of the union (the total space covered by all three forests) of the NROY spaces. The "initial" space represents the total parameter space covered by the ensemble.

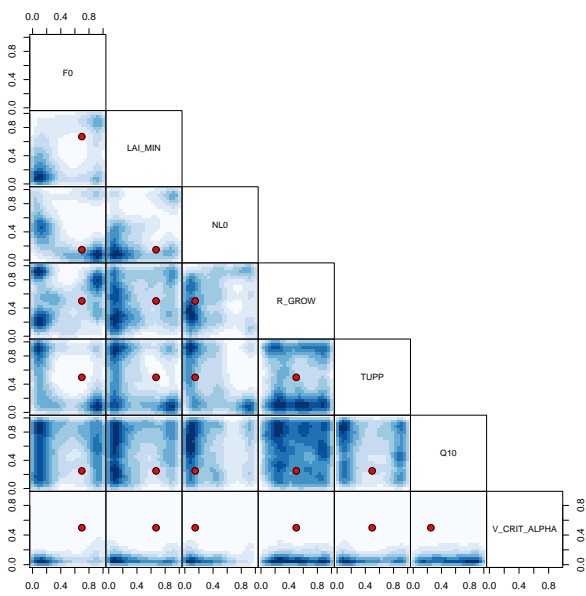

**Figure 11.** NROY land surface input space shared by all three forests before bias correction. Blue shading denotes the density of NROY input candidates, projected into the two dimensional space indicated by the labels. The default parameter settings are marked as red points.

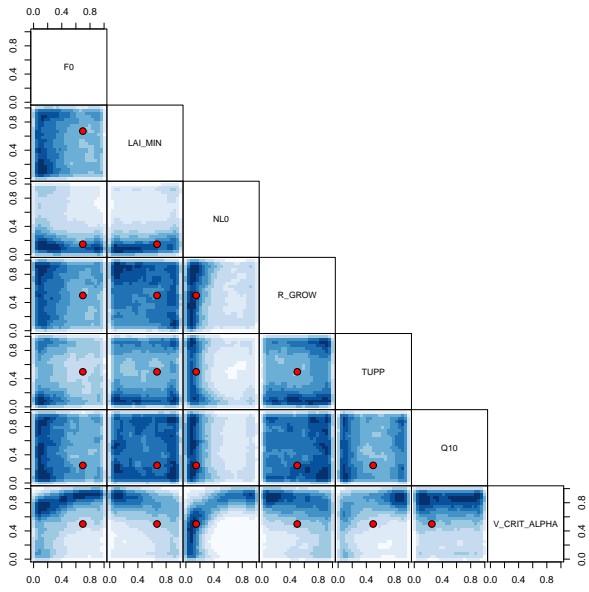

**Figure 12.** NROY land surface input space shared by all three forests when bias corrected using the augmented emulator. Blue shading denotes the density of NROY input candidates, projected into the two dimensional space indicated by the labels. The default parameter settings are marked as red points.

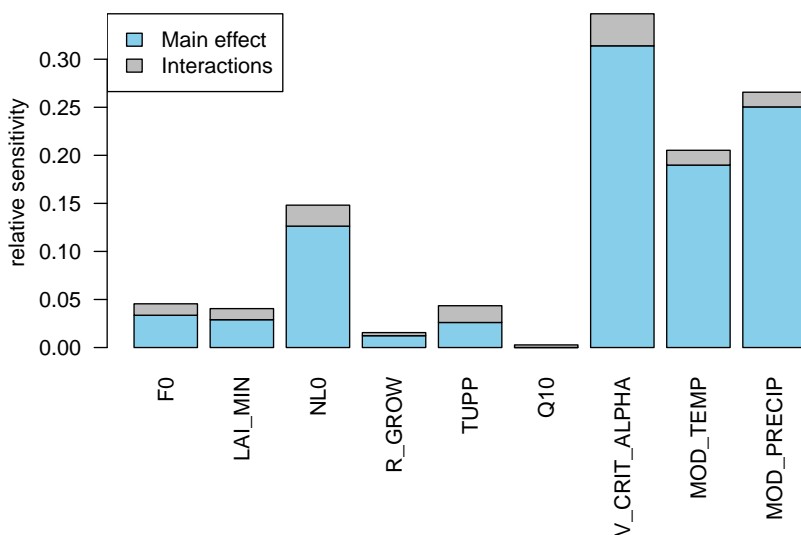

**Figure 13.** Sensitivity of forest fraction to model parameters and climate parameters, found using the FAST99 algorithm of Saltelli et al. (1999).

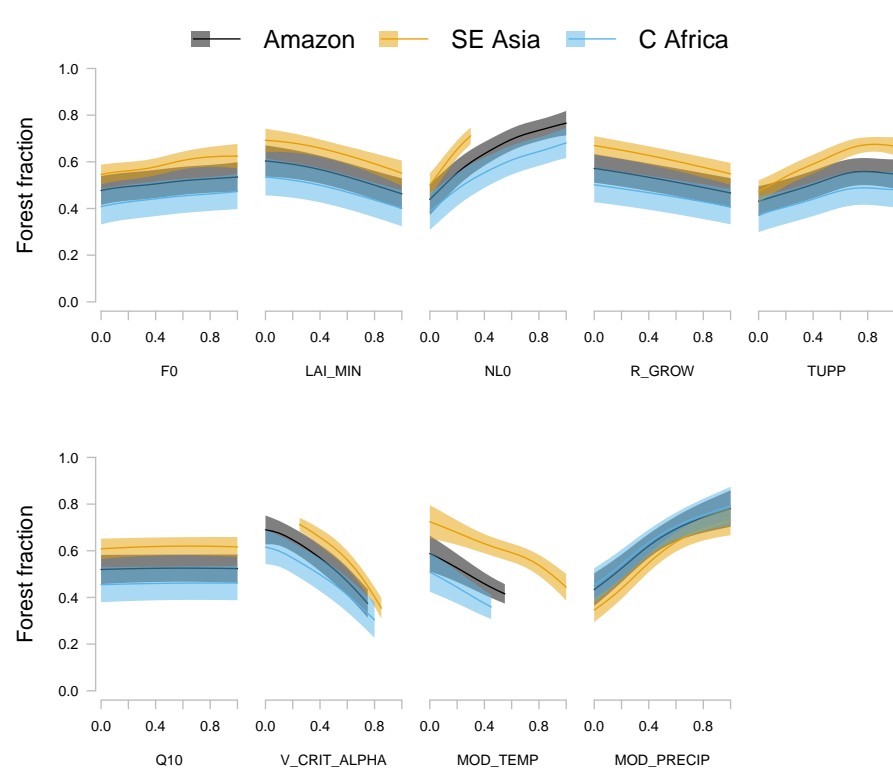

**Figure 14.** One-at-a-time sensitivity of forest fraction variation of each parameter and climate variable in turn across the "Not Ruled Out Yet" parameter range. All other parameters or variables are held at their default values while each parameter is varied, and values of model broadleaf forest fraction which are statistically far from observations are excluded. Solid lines represent the emulator mean and shaded areas represent ± 2 standard deviations of emulator uncertainty.

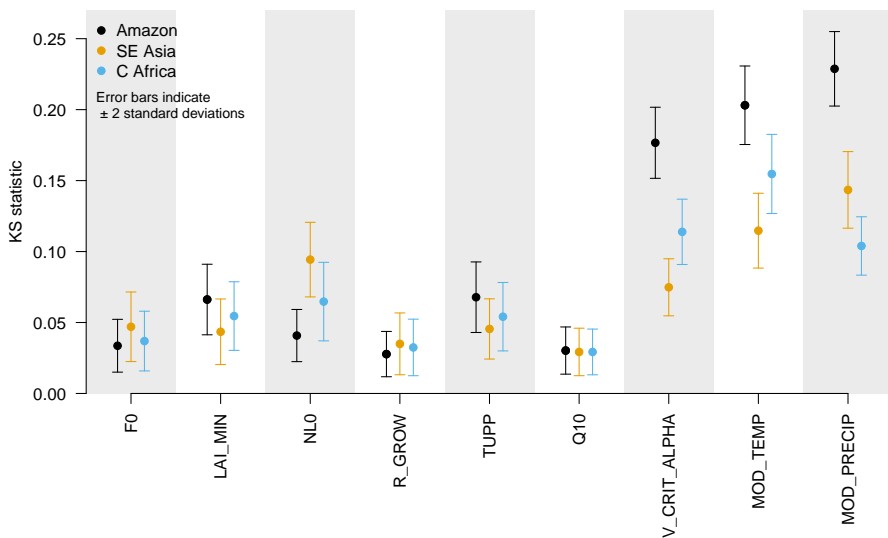

**Figure 15.** Monte Carlo filtering estimate of the sensitivity of model output to inputs, using 5000 emulated members. Error bars represent ± 2 standard deviations of uncertainty in the statistic, calculated by repeating the calculation 1000 times.

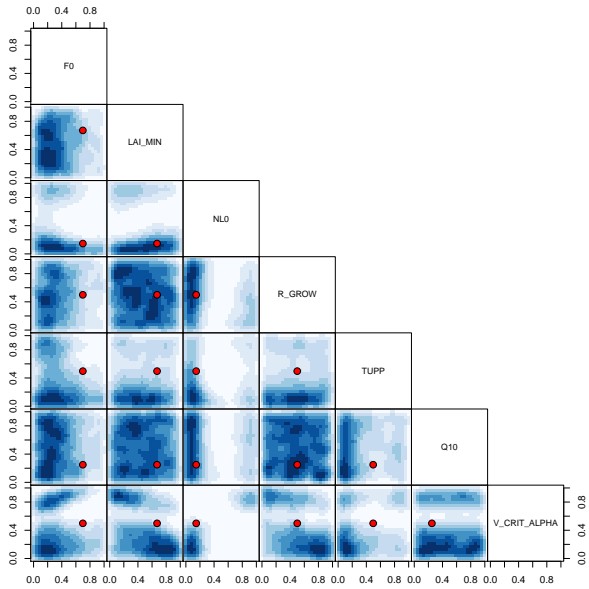

**Figure 16.** Two dimensional projections of the density of inputs where the corresponding bias corrected emulated forests have a smaller error than the bias corrected default parameters. These regions might be good targets for additional runs of the climate model. Default parameters are shown as a red point.

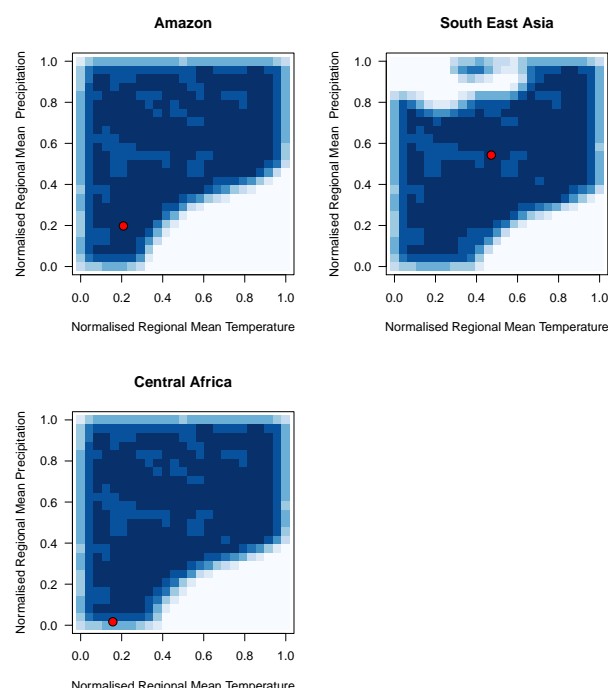

**Figure 17.** Density of not-ruled-out yet emulated temperature and precipitation pairs for each observed tropical forest fraction, when input parameters are held at their default values. Observed climates for each forest are marked in red.