# Peer review of "Correcting a bias in a climate model with an augmented emulator"

_Geoscientific Model Development, 2019_

## Referee Comment (RC1) · Anonymous Referee #1 · 12 Sep 2019

**Paper:** Correcting a bias in a climate model with an augmented emulator

**Authors:** McNeall *et al*.

**Review:**

In this paper the authors present a technique to determine the set of plausible parameter combinations that lead to a credible version of the climate model FAMOUS, in the presence of a significant model bias.

This study follows on from the work described in McNeall *et al*. (2016), using the same perturbed parameter ensemble of model runs from the FAMOUS climate model. McNeall *et al*. (2016) identified the land surface input space of FAMOUS that was consistent with observations of the output 'forest fraction' for different forest regions, and found that there was very little overlap between parameter space that was consistent with observations of the Amazon region and parameter space that was consistent with observations of the other forest regions. This led them to conclude that there is either a local climate bias and/or some missing/incorrect process(es) in the land surface model of FAMOUS.

Here, the authors extend the approach of McNeall *et al*. (2016) to develop a method that accounts for a climate bias in the model-observation comparison, by bias-correcting emulator-predicted model output before it is used in the statistical 'history matching' procedure that determines the plausibility of each tested realisation (combination of land surface model parameters) of FAMOUS . The authors bias correct the climate of the Amazon using an 'augmented' Gaussian process emulator, where temperature and precipitation outputs are treated as model inputs alongside the uncertain land surface input parameters. They find that the forest fraction in a region is sensitive to these climate variables, and by bias correcting the climate in the Amazon region the authors are able to correct the forest fraction in the Amazon to tolerable levels for many of the tested realisations of FAMOUS, including the default parameter set, thus increasing the amount of valid input space shared with the other forest regions (from 1.9% of the parameter space in McNeall *et al*. (2016) to 28.3% here).

This approach is a novel adaptation to current methods for climate model-observation comparison, which shows potential for improving and simplifying the model tuning process for coupled climate models, as well as aiding in the identification of model errors. The manuscript is well-written and definitely falls within the scope of GMD and EGU. I like the concept of the 'augmented emulator' that includes the localised climate outputs as inputs, however, I do have some concerns about the true validity of the augmented emulator, in particular for the bias corrected climates of central Africa and to an extent the Amazon in the application (see Specific comments below). If these concerns, along with the other comments listed, can be addressed then I would recommend the publication of the manuscript in GMD.

**Specific Comments:**

– **Page 7 Line 11 – Page 8 Line 2:** I'm confused by the role of the 'beta' parameter in the ensemble set-up. I don't think I understand this. The simulations in the ensemble have each been run with one of 10 different configurations of the atmosphere? Was it randomly assigned as to which 'atmosphere configuration' each simulation had? What are the implications of this? Doesn't this introduce biases into the ensemble (into the ensemble outputs) for identifying the parameter combinations that are plausible, if the ensemble members do not have the same starting point in the atmospheric set-up? Please clarify this in the text.

– **Page 8 Line 7-8; Section 3.1:** 'The study only considers regions dominated by tropical broadleaf forest, so as not to confound analysis by including other forests which may have a different set of responses to

perturbations in parameters, rainfall and temperature'. Even though the forest regions are of the same 'type' (tropical broadleaf forest), are there other factors in the model that might affect the forest response between regions that are not accounted for? Such as topography that might affect how the forests respond to the parameter perturbations?

The analysis is based on an assumption that the forests in the different regions will have the same responses to changes in the land surface parameters, rainfall and temperature. How realistic is this assumption? I'm not saying this assumption should not be made (we have to make assumptions for modelling and statistical analysis!), but I think the authors should state more clearly (in section 2 or 3) that they make this assumption (it is implied in the set-up of the augmented emulator, but not openly said), and discuss any possible implications of it on the results. This could be a further reason why the other forests 'do slightly less well' (e.g. Page 16, Line 11; Page 13 Line 12; Table 2).

– **Section 3.2 – validation of the augmented emulator:** The augmented emulator is validated using a 'leave-one-out' approach. However, I am not convinced that this approach fully validates the emulator for its use in the following analysis in Section 4. The emulator looks to be sampled from beyond the range of its training data where it is not validated, which could be affecting the results obtained.

The augmented emulator is only trained to predict the forest fraction for climates (P and T variable combinations) that occur in the original simulations of the ensemble for the 3 regions. Figure 3 shows that coverage of the climate variables [P,T] 2-dimensional state-space is not uniform, and has sparse (if any) coverage in many areas of that 2-d space. In particular, there is rather limited coverage around the 'observed' climate (P,T combination) for the Amazon, and for Central Africa there looks to have no training points particularly close by. How the information for P and T augment on to the 7-dimensional space-filling design of the land-surface parameters to produce the final 9-dimensional input design with which the emulator is constructed is not shown (I imagine the actual coverage is some weird and complicated shape with some potentially large gaps) and so I wonder what the training data coverage, and hence emulator skill, is like for the areas of that 9-dimensional parameter space that are used in the bias correction analysis for these observed climates? The leave-one-out validation approach is only testing the emulator in the areas of space that have training runs, and so although the validation plots in Section 3.2 seem reasonable, it looks to me that the emulator is not tested (and so not validated) for the 'bias- corrected' climate of central Africa, and the Amazon, where the emulator is densely sampled for the analysis in Section 4. Outside the trained area of parameter space (e.g. where the observed climate for Central Africa is) prediction from the emulator becomes extrapolation from the emulator, the emulator prediction uncertainty can quickly increase and prediction values from the emulator will return back towards the form of the prior specification of the mean functional form from the GP emulator construction (the emulator mean response surface will bend/shape back towards that form), here a linear function of the inputs [stated in the supplementary information]. Hence, how do we know that the bias corrected predictions used in the analysis for these regions are sensible?

Can the authors provide some further validation for the emulator predictions for the observed climates of central Africa and the Amazon? If not, then the authors should explicitly state this limitation of the emulator in the paper and discuss the possible consequences of this on the figures and results presented in Section 4, and in the discussion and conclusion Sections 5 and 6.

Also, could the emulator predicted responses to climates not covered by the training data (including the observed climate for Central Africa and the top left area in Fig 3) be dependent on the emulator's prior (linear) form, and change if this specification was changed? If yes, how confident are the authors in the prior emulator form (linear) being representative of the climate model's actual behaviour in parameter space beyond the training data? If they are not confident in it, then it either shouldn't be used or it needs to be more carefully specified so that it can be used.

In particular:

- The results shown in Fig 8 are obtained by sampling with the climate variables set at the observed values (shown in Fig 3). Hence, this means that for central Africa (and the Amazon to some extent), the sampled predictions come from extrapolating from the emulator beyond the extent of the training data. The responses to the climate variables are reasonably linear and I wonder if this response is at least partially driven by the form of the prior specification of the GP emulator mean function?

- Are the results in Fig 9 from the FAST99 algorithm generated by sampling the across the full 2-dimensional climate [P,T] space? How are these results affected by sampling from the emulator where there is no training data, particularly for a cool/wet climate at the top left of Fig 3? Could the sensitivity to the climate variables be over-estimated here? (I cannot easily tell from the plot, but do the individual main effect sensitivities sum to <1? (Main effect + interaction should sum to 1.) I've seen instances where the algorithm produces main effect values that sum to >1 in the presence of noise in the emulator fit.)

- In Figure 10 the emulator is used to simulate across the entire range of simulated temperature and precipitation with all other inputs fixed at the default setting. How might this result be affected?

- How might this issue affect the results of the retained parameter space from the history matching (Figures 13-16) for each forest region? In Figure 16, are the regions not covered by the training data more likely to be retained as emulator error is larger, reducing the value of the implausibility metric so that it cannot be ruled out?

— **Page 15 Line 3-4, Figure 15 (and discussion):** guiding further runs, choosing high density regions to run new ensemble members. This is a useful outcome. My question here really relates to how the results trace back to improve model performance... How does the bias correction information feed back for the modeller to know what 'atmospheric configuration' should be used (the 10 atmospheric parameters, or beta?) with the inputs selected as good for any new runs? Can a good representation of forest fraction for all forests simultaneously be obtained in new runs at these parameter combinations without bias correction? Or, would any new runs always need to be bias corrected too, until further work to understand the true cause of the climate bias is completed and the climate model updated? As obviously, just running the model at more combinations in this identified joint space will induce climates as shown in Figure 3, away from the observed climates for each forest. Could the authors comment on this in the discussion?

**Further Comments:**

— **Page 3 Line 18:** 'Without strong prior information...' What is meant by 'prior information' here? What kind of information? On observations? On model skill? On both? This is a bit vague and needs more clarity.

— **Page 5 Line 9:** What reasons? Please give more details here: '...whereas there were a number of reasons one might reject the proposed parameter space, **including...**'

— **Section 1.3 (Page 5 Line 11):** It might give more context to the first listed aim on line 11, and for the detail coming in the second paragraph of the section (discussing the results of McNeall et al (2016), which are not 'aims of this paper') to connect that this study is extending the analysis of McNeall et al. (2016) at the start of the section in the first line (first aim?)?

- **Page 6 Line 9-11:** Sentence starting 'Parameter perturbations...'. Are there any references for examples of such findings?

- **Page 6 Line 19:** '...was sensitive to perturbations in parameters,'. This is vague... What kind of parameters? Edit to say '...was sensitive to perturbations in parameters **such as**...'

- **Page 6 Line 31 – Page 7 Line 1:** I realise that the details of the ensemble are in McNeall et al (2016), but I think it would be useful to give minimal details of the parameters perturbed in this paper also. Their effects are being compared to the climate variables in Section 4.1, with parameter names (acronyms) given in the text, and yet I have to go to a completely different paper to find a description of them /what they correspond to in the model. Please add a small summary table (to the supplementary file, if not to the main paper) that lists the parameters with short descriptions.

- **Page 7 Line 4:** What is meant by 'global values'? Global values of what? Please clarify.

- **Page 8 Line 30:** Please provide a reference for GP emulation.

- **Page 9 Line 13:** '...the 10 atmospheric parameters perturbed in a previous ensemble, summarised by the β parameter'. I'm struggling to picture how the effects of 10 parameters can be summarised by 1 parameter. A lot of information is being condensed here? This needs more explanation. (Also see first comment above under 'Specific comments'.)

- **Page 9 Line 14:** This sentence: 'We cannot control them directly and thus ensure that they lie in a latin hypercube configuration' is confusing and could be interpreted in different ways. I first read 'and thus ensure' as that you **do** ensure that they lie in a LH configuration. But on second read I see the meaning you want is that you can't ensure this. Please re-phrase.

  Also, the 'latin' in 'Latin hypercube' should have a capitol L. Please update here and elsewhere.

- **Page 10 Line 12:** '3% of the maximum possible value of the ensemble.' What does this really statement tell us? The maximum forest fraction is 1 so the error of 0.03 is 3% of this forest fraction value, but this is the minimum percentage of an output value that it could be. The majority of predictions will be less than 1, and so the error of an 'average prediction' is in general a larger percentage than this. It seems a bit of a misleading statement, and I suggest removing it here and on line 14.

- **Page 10 Line 25:** I don't understand how the 'rank histogram' indicates that we have 'reliable' uncertainty estimates? It shows the predictions are a mixture of over and under-estimations of the actual model, but it gives no indication of the size of errors, which could be large and therefore not reliable? Maybe I misunderstand this.

- **Section 4.2 and Figure 10:** The interpretation of this plot needs clarification. Would the contours be similar at different parts of the land-surface parameter space? (Fixing at a different simulation to the default?) On Page 11, Line 26, it suggests that for central Africa, moving any ensemble member (small triangle point) to the observed (big triangle) would not cross many contours, but the central Africa points with wetter climates (normalised T at approx. 0.4, normalised P at approx. 0.5 to 0.6) would cross between 3 and 4 contours to be in the same one as the observed (big triangle), so I'm not sure this is true? Please clarify this.

- **Page 12 Line 8-11:** Are the values given in this paragraph mean absolute error between model and observations? The first line says 'difference', but this must be an average? Please clarify. Also, could the lack of training data near the observed climate for central Africa be a contributing factor to why central Africa is worse (Line 9) in this metric?

- **Section 4.4:** It might be better for the flow of the results section if the first part of Section 4.4 (up to Page 13 Line 2) describing the history matching methodology was moved into Section 3 on methods?

- **Page 13 Line 12:** Could more detail be given as to why the implausibility value at the default settings rises for central Africa and SE Asia on bias correction?

- **Page 17 Line 14-16:** Summarising the 10 parameters using 2 outputs is useful, but this is likely to have little traceability back to the original 10 input values? Many different combinations of the 10 inputs could lead to a similar combination in the 2 outputs, so how would one know what combination of the 10 inputs is best when setting up any further runs of the model? Also, the '*O(10xp)*' rule is when training points are space-filling across the parameter space being emulated. There is no guarantee (as seen in this example) that this property will hold when outputs are used as dimension reduction, so more points, or even less points, could easily be required. This should be acknowledged.

**Technical corrections:**

- **Page 1 Line 13-14:** 'This might be due to...'.  I think this sentence might be easier to read if the number/list format is replaced with 'This might be due to either ..., or...., or a combination of both.'

- **Page 1 Line 16-17:** '...alongside **regular** land surface input parameters.'.  Is the term 'regular' needed here? Maybe remove this word.  [The 'regular' suggests to me that there may be other types of land surface input parameters that are 'not regular' which are not included, which I don't think is the case.]

- **Page 1 line 15:** Should '...a climate model...' be '...**the** climate model...'?  This is now the specific climate model used by McNeall et al (2016).

- **Page 1 Line 17-18:** For readability, please change 'is nearly as sensitive to climate variables as changes in **its** land surface parameter values.' to 'is nearly as sensitive to climate variables as **it is to** changes in land surface parameter values.'

- **Page 2 Line 9:** Should '...processes sufficiently to trust...' be '...processes sufficiently **and** to trust...'?

- **Page 2 Line 28:** The sentence here is hard to read. Change: '...practices, there appear no standard procedures for climate model tuning however - as the authors...' to '...practices, there appear **to be** no standard procedures for climate model tuning. **H**owever**,** as the authors...'.

- **Page 2 Line 32:** Missing word? Edit: 'It might start with single column version...' to be 'It might start with **a** single column version...'

- **Page 3 Line 2:** Change word order? Edit: '...might be then tuned...' to '...might **then be** tuned...'

- **Page 3 Line 8:** Change 'Golaz et al. (2013) Show...' to 'Golaz et al. (2013) **s**how...'.

- **Page 3 Line 20-21:** This sentence needs plural 'candidates' at the start, and the second part should be given as more of a negative to make the point? Revise to: 'This means that **good candidates** for input parameters might be found in a large volume of input space, **but** projections of the model made with candidates from across that space might **diverge to** display a very wide range of outcomes.'

- **Page 3 Line 23:** 'individual parts' Of the tuning process? Or the model? Please clarify.

- **Page 4 line 14:** Missing full stop after Vernon et al. (2010).

- **Page 4 Line 20, Line 23:** References in bracketed format when should be in in-line format.

- **Page 4 line 33:** Remove the second 'used'.

- **Page 5 line 1:** Missing words. Change to: '...structural bias in **the** ocean component of **the** climate model HadCM3 could'

- **Page 5 Line 33:** Should this be '...use the **augmented** emulator to estimate the sensitivity...'?

- **Page 7 Line 2:** Move the reference to Fig 1 to the next sentence, which is the sentence that is describing what is shown in Fig 1.

- **Page 7 Line 7-8:** '...parameter settings which the **emulator** suggested should lead to **an** adequate simulations of...'. The emulator provides the predictions of model output but does not indicate adequacy – this comes from the history matching process (as the authors have described). Change to: '...parameter settings which the **history matching process** suggested should lead to adequate simulations of...'.

- **Page 8 Line 9, 12:** References to Jones et al, and Adler et al are in in-line format when should be in bracketed format?

- **Page 9 Line 10:** Change to: '...each of the forests**:** the Amazon, central Africa and Southeast Asia...' or put the forest region names in brackets?

- **Page 9 Line 11-12:** For readability, move the sentence 'Regional extent of ... supplementary material.' so that it is the second sentence in this paragraph. (The next sentence follows better from the one before it!)

- **Page 9 Line 24:** Here the work by M16 is referred to as being by the authors of this work ('we built'), but in all previous references to this point it has been referred to as a separate study (e.g. M16 argue..., or M16 speculated...). Update as needed to be consistent.

- **Page 11 Line 3:** Should this paragraph start with: 'The **augmented** emulator...'

- **Page 11 Line 5:** '...predict changes in forest fraction as each variable is changed from the lowest to highest setting in turn...'. Should 'variable' in this sentence be replaced with 'input'? – as this is done for the land surface input parameters as well as the climate variable inputs?

- **Page 12 Line 22:** Remove the second 'the'.

- **Page 13 Line 21:** The term 'the climate-bias forest' sounds weird? Should this be 'the climate-bias-corrected forest'?

- **Page 16 Line 34:** Remove the second 'could'.

- **Page 17 Line 14:** '$O(170)$' should be in italics?

- **Page 18 Line 6:** Change: 'If trained an ensemble...' to 'If trained **on** an ensemble...'. Also, remove the second 'which'.

- **Page 18 Line 10:** Remove; '(e.g.)'.

- **Page 18 Line 19:** Should 'learned' be 'learn'?

- **Page 18 Line 32:** Missing word. Change: '...in leave-one-out...' to '...in **a** leave-one-out...'.

- **Page 19 Line 13:** Change 'finding' to 'finding**s**'.

- **Page 27, Fig 5 caption:** The brackets for $g_1$ are in the wrong place? For $g_n$ and $y_n$, should 'n' be replaced with '3', as is shown in the diagram, and written for 'C' in the next sentence?

- **Supplementary information Line 17:** Reference in bracketed format when should be in in-line format.

- **Supplementary information Line 18:** Missing word. Change: '...in section 3 the...' to '...in section 3 **of** the...'

---

## Referee Comment (RC2) · Anonymous Referee #2 · 28 Oct 2019

The authors seek to produce an emulator that can account for some of the known biases in a climate model when the aim is to find a 'good' parameter set to represent observations. Overall, I think the idea is a good one and the augmenting of the emulator in this case clearly works to make a better emulator for model constraint. The paper is well written and the method easy to follow. The work should be published in GMD.

I have a few points to discuss:

I am pretty confused about the 'beta' parameter and what it means in both the original model and the emulator here – can this be clarified in the text please.

How much do you need to know about the present bias? It's clear the authors had information on this and a good idea from the modellers where the biases came from

but it's less clear what they may have done with less information.

Page 11,Line 26: 'Moving any . . .' I find this sentence confusing. Can you better link it to the figure and clarify?

Figure 14: There is a clear relationship with V_CRIT_ALPHA and NLO when the augmented emulator is used. Can you discuss this and what it might mean?

Page 17, line 5: Is it temperature and precipitation that should be targeted or how the model treats them? It's not clear to me exactly what you are recommending.

Page 17, line 18: If you were to this for every grid box would you expect predictability?

I could see this method working for elevation and seasonal biases? How might you go about this?

---

## Author Comment (AC1) · 7 Feb 2020

**Author response to reviewers for "Correcting a bias in a climate model with an augmented emulator" by McNeall et al. (2019)**

We thank the reviewers for their constructive, fair, and thorough review. In this response, reviewer comments are in black text and author responses are in blue text. Several of the figures used in this response to reviewers have been added to the supplementary material, or to the main text. We identify these with dual labels outlining their position in this response, and in the updated manuscript and supplementary material. Responses to both reviewers are contained in this supplement, and a list of references not used in the main text can be found at the end.

**Reviewer 1**

In this paper the authors present a technique to determine the set of plausible parameter combinations that lead to a credible version of the climate model FAMOUS, in the presence of a significant model bias. This study follows on from the work described in McNeall et al. (2016), using the same perturbed parameter ensemble of model runs from the FAMOUS climate model. McNeall et al. (2016) identified the land surface input space of FAMOUS that was consistent with observations of the output 'forest fraction' for different forest regions, and found that there was very little overlap between parameter space that was consistent with observations of the Amazon region and parameter space that was consistent with observations of the other forest regions. This led them to conclude that there is either a local climate bias and/or some missing/incorrect process(es) in the land surface model of FAMOUS. Here, the authors extend the approach of McNeall et al.(2016) to develop a method that accounts for a climate bias in the model-observation comparison, by bias-correcting emulator-predicted model output before it is used in the statistical 'history matching' procedure that determines the plausibility of each tested realisation (combination of land surface model parameters) of FAMOUS. The authors bias correct the climate of the Amazon using an 'augmented' Gaussian process emulator, where temperature and precipitation outputs are treated as model inputs alongside the uncertain land surface input parameters. They find that the forest fraction in a region is sensitive to these climate variables, and by bias correcting the climate in the Amazon region the authors are able to correct the forest fraction in the Amazon to tolerable levels for many of the tested realisations of FAMOUS, including the default parameter set, thus increasing the amount of valid input space shared with the other forest regions (from 1.9% of the parameter space in McNeall et al. (2016) to 28.3% here).This approach is a novel adaptation to current methods for climate model-observation comparison, which shows potential for improving and simplifying the model tuning process for coupled climate models,as well as aiding in the identification of model errors. The manuscript is well-written and definitely falls within the scope of GMD and EGU.I like the concept of the 'augmented emulator' that includes the localised climate outputs as inputs, however, I do have some concerns about the true validity of the augmented emulator, in particular for the bias corrected climates of central Africa and to an extent the Amazon in the application (see Specific comments below). If these

concerns, along with the other comments listed, can be addressed then I would recommend the publication of the manuscript in GMD.

**Specific Comments:**

Reviewer 1
Page 7 Line 11 – Page 8 Line 2: I'm confused by the role of the 'beta' parameter in the ensemble set-up. I don't think I understand this. The simulations in the ensemble have each been run with one of 10 different configurations of the atmosphere? Was it randomly assigned as to which 'atmosphere configuration' each simulation had? What are the implications of this? Doesn't this introduce biases into the ensemble (into the ensemble outputs) for identifying the parameter combinations that are plausible, if the ensemble members do not have the same starting point in the atmospheric set-up? Please clarify this in the text.

Author response
The reviewer is correct the simulations in the ensemble have each been run with one of 10 different configurations of the atmosphere and ocean. The beta parameter indexes each of these sets, with the lowest values of beta being the best performing according to Gregoire et al. (2010). The performance of FAMOUS was evaluated in that paper using a wide range of model outputs, and so the "best performing" parameter sets are not necessarily the best performing in temperature and precipitation over the Amazon region, and are drawn from a relatively large region of input parameter space.

The fact that atmospheric and oceanic parameters are also perturbed does indeed have implications for identifying the (land surface and vegetation) parameter sets that are plausible, as the climate of the system - important for land surface behaviour - is affected. Correcting these potential biases in climate is the aim of the augmented emulator, effectively using temperature and precipitation model output to summarise perturbations across the 10 atmospheric and oceanic parameters perturbed in the previous ensemble. We have made changes to the text in section 2.1 "Biases in FAMOUS" to clarify this point, and the explanation of the beta parameter has been expanded. The updated paragraph now reads:

"The ensemble of 100 members perturbed 7 land surface and vegetation inputs (see supplementary material, table S1 along with a further parameter denoted "beta" (β). Each of the ten values of beta provides an index to one of ten of the best-performing atmospheric and oceanic parameter sets used in a previous ensemble with the same model (Gregoire et al 2010) with the lowest values of beta corresponding to the very best performing variants. The beta parameter therefore summarised perturbations in 10 atmospheric and oceanic parameters that impacted the climate of the model, randomly varied with land surface input parameters, and potentially leading to different climatologies in a model variant with the same land surface parameters but different values of beta. Variations in the beta parameter did however not correlate strongly to variations with any of the oceanic, atmospheric or land surface parameters in the ensemble, and so the parameter was excluded from the analysis in M16. In this analysis we recognise that the different model climates caused by variations

in the atmospheric and oceanic parameters will have an impact on the forest fraction, and so we summarise those variations directly using local temperature and precipitation."

Reviewer 1
Page 8 Line 7-8; Section 3.1:'The study only considers regions dominated by tropical broadleaf forest, so as not to confound analysis by including other forests which may have a different set of responses to perturbations in parameters, rainfall and temperature'. Even though the forest regions are of the same 'type' (tropical broadleaf forest), are there other factors in the model that might affect the forest response between regions that are not accounted for? Such as topography that might affect how the forests respond to the parameter perturbations? The analysis is based on an assumption that the forests in the different regions will have the same responses to changes in the land surface parameters, rainfall and temperature. How realistic is this assumption? I'm not saying this assumption should not be made (we have to make assumptions for modelling and statistical analysis!), but I think the authors should state more clearly (in section 2 or 3) that they make this assumption (it is implied in the set-up of the augmented emulator, but not openly said), and discuss any possible implications of it on the results. This could be a further reason why the other forests 'do slightly less well' (e.g. Page 16, Line 11; Page 13 Line 12; Table 2).

Author Response
The reviewer makes a good point. We have added two paragraphs to the main text. At the end of section 2.1 "Biases in FAMOUS", we have added:

"We are assuming here that tropical forests can be represented by a single set of forest function parameters. Whilst such an assumption risks missing important differences across heterogeneous tropical forests, modelling the system with the smallest set of common parameters avoids overfitting to present day data. Avoiding overfitting is important if we are to use these models to project forest functioning in future climates outside observed conditions. One of the questions that the analysis presented in this paper addresses is whether current forest biases in the simulations reflect limitations of this single tropical forest assumption, or whether biases in the simulations of the wider climate variables play a more important role."

We have added a paragraph at the end of section 3.1 "An augmented emulator", explicitly outlining the assumptions of the augmented emulator, and briefly highlighting what might happen if such an assumption was unjustified:

"We note that the augmented emulator depends on the assumption that modelled broadleaf forests in each location respond similarly to perturbations in climate and input parameters. This assumption may not hold for the behaviour of the forests in the model, or indeed the real world. For example, particularly deep rooting of forests in the Amazon would respond differently to rainfall reductions but these processes are not represented in the underlying climate model. Similarly, differing local topology that is captured in the climate model, may influence the forests in a way not captured by our emulator. In both cases, the emulator would show systematic errors of prediction."

Reviewer 1

Section 3.2 –validation of the augmented emulator: The augmented emulator is validated using a 'leave-one-out' approach. However, I am not convinced that this approach fully validates the emulator for its use in the following analysis in Section 4. The emulator looks to be sampled from beyond the range of its training data where it is not validated, which could be affecting the results obtained.The augmented emulator is only trained to predict the forest fraction for climates (P and T variable combinations) that occur in the original simulations of the ensemble for the 3 regions. Figure 3 shows that coverage of the climate variables [P,T] 2-dimensional state-space is not uniform, and has sparse (if any) coverage in many areas of that 2-d space. In particular, there is rather limited coverage around the 'observed' climate (P,T combination) for the Amazon,and for Central Africa there looks to have no training points particularly close by. How the information for P and T augment onto the 7-dimensional space-filling design of the land-surface parameters to produce the final 9-dimensional input design with which the emulator is constructed is not shown (I imagine the actual coverage is some weird and complicated shape with some potentially large gaps) and so I wonder what the training data coverage, and hence emulator skill, is like for the areas of that 9-dimensional parameter space that are used in the bias correction analysis for these observed climates? The leave-one-out validation approach is only testing the emulator in the areas of space that have training runs, and so although the validation plots in Section 3.2 seem reasonable, it looks to me that the emulator is not tested (and so not validated) for the 'bias-corrected' climate of central Africa, and the Amazon, where the emulator is densely sampled for the analysis in Section 4. Outside the trained area of parameter space (e.g. where the observed climate for Central Africa is) prediction from the emulator becomes extrapolation from the emulator, the emulator prediction uncertainty can quickly increase and prediction values from the emulator will return back towards the form of the prior specification of the mean functional form from the GP emulator construction (the emulator mean response surface will bend/shape back towards that form), here a linear function of the inputs [stated in the supplementary information]. Hence, how do we know that the bias corrected predictions used in the analysis for these regions are sensible?

Can the authors provide some further validation for the emulator predictions for the observed climates of central Africa and the Amazon? If not, then the authors should explicitly state this limitation of the emulator in the paper and discuss the possible consequences of this on the figures and results presented in Section 4, and in the discussion and conclusion Sections 5 and 6. Also, could the emulator predicted responses to climates not covered by the training data (including the observed climate for Central Africa and the top left area in Fig 3) be dependent on the emulator's prior (linear) form, and change if this specification was changed? If yes, how confident are the authors in the prior emulator form (linear) being representative of the climate model's actual behaviour in parameter space beyond the training data? If they are not confident in it, then it either shouldn't be used or it needs to be more carefully specified so that it can be used.

In particular:

The results shown in Fig 8 are obtained by sampling with the climate variables set at the observed values (shown in Fig 3). Hence, this means that for central Africa (and the Amazon

to some extent), the sampled predictions come from extrapolating from the emulator beyond the extent of the training data.The responses to the climate variables are reasonably linear and I wonder if this response is at least partially driven by the form of the prior specification of the GP emulator mean function?

Author response
The reviewer raises an important point about emulator extrapolation. We hope that the following further validation of the emulator and exploration of the importance of the prior emulator specification will allay the reviewer's concerns, and that any remaining concerns are sufficiently addressed in the text. We have added section 2 "Further validation" to the supplementary material, with the indicated figures from this response included.

The reviewer is concerned first that the design points do not offer enough coverage near the observed temperature and precipitation for Central Africa and the Amazon, in particular that the emulator is forced to extrapolate, and may be poor at these locations. Further, given that the emulator is extrapolating, the reviewer is concerned that the prior form of the emulator may be dominant, and the prediction overly dependent on that prior form.

In an ideal world, we would generate ensemble members at or near the observations in question, as a way to validate the emulator and ensure our predictions are correct. This is impractical for two reasons 1) we don't have access to the model and setup in order to generate new runs. While it sounds like a weakness of the design, this is a feature of the paper, in that this is a common situation when analysts are working with models from other groups, with older versions of the model, or with very computationally expensive models where more runs cannot be afforded. 2) There is no way to directly control the temperature and precipitation in the model in order to generate a particular design. These inputs to the emulator are in fact outputs of the model, controlled largely by an inaccessible set of parameter perturbations. Given that we cannot validate the emulator at the observations, we suggest that we can at least show that the emulator performs well, even when required to extrapolate into the broader region of temperature and precipitation where the observations in question lie.

Our initial emulator validation relied on a leave-one-out-metric, but we are able to hold out a larger sample of ensemble members, and check the predictions of the resulting emulator. In this case, we hold out 6 ensemble members in the region of and nearest to the observations of temperature and precipitation of the Amazon and Central Africa. We hold out ensemble members with a precipitation below 0.2 and temperature below 0.4 in the normalised ensemble. These ensemble members occur in the bottom-left of the temperature-precipitation phase space, closest to the Central African and Amazon observations (figure R1, figure S1 in the supplementary material). They consist of three members each from the Central African and Amazon forests. These held-out members include one member at the very edge of the temperature space, that is it must be a marginal extrapolation. In our experience, marginal extrapolation is less accurate than extrapolating within the marginal limits of a multidimensional space. We therefore test the emulator with a much more challenging prediction than the leave-one-out validation.

[Figure]

Figure R1/S1. Location of the held-out ensemble members (red points), used to test extrapolation of the emulator.

Figure R2 (also figure S2 in the supplementary material) shows the prediction of the 6 held-out ensemble members (red dots) in the context of the leave-one-out validation (grey dots). In the held-out case, we fit the emulator based on the 294 remaining ensemble members, and predict all 6 held out members at the same time. As the training set is slightly smaller than each leave-one-out training set (299 members), and the emulator is expected to extrapolate further, we might expect a significant degradation in the performance of the emulator in prediction. As we see in figure R2/S2, there is little evidence of such a degradation. Both prediction error and estimated uncertainty are well within the bounds of that found during the leave-one-out validation exercise. Figure R3 (figure S3 in the supplementary material) shows the prediction error for the 6 members, in the context of the histogram of errors from the leave-one-out exercise. None of the errors are near the limits of the distribution, even though they might be expected to be larger, with a smaller training set and deeper extrapolation.

[Figure]

Figure R2/S2. Predictions of forest fraction ensemble members in a leave-one-out validation exercise (grey dots) and for the 6 held-out ensemble members (red dots). Vertical lines represent ± 2 standard deviations.

[Figure]

Figure R3/S3. Emulator prediction error (red rug plot) of 6 held-out ensemble members in the context of the leave-one-out validation exercise (grey histogram).

When making a direct comparison of prediction of the 6 held-out members (figure R4 and figure S4 in the supplementary material), we see that there is some small degradation in the performance of the emulator - predictions tend to be slightly further from the held-out ensemble member, and uncertainty bounds wider. However, it should be noted that the error of the held-out samples is 1) only slightly larger than in the leave-one-out case, 2) small when compared to the range of the ensemble, and 3) prediction uncertainty intervals are certainly appropriate and do not increase dramatically. There seems to be no question that even when asked to predict ensemble members that are near the edge of parameter space, and are a significant extrapolation, the emulator performs well. Obviously, this shouldn't be taken as meaning that there is no risk of the emulator performing poorly when extrapolating to the regions of the Amazon and central African temperature and precipitation. However, we hope we have shown that there is little evidence to suggest that the emulator will perform poorly there.

[Figure]

Figure R4/S4. Direct comparison of prediction of the held-out ensemble members in both the leave-one-out (LOO, blue points) and held-out (red points) validation exercises. Vertical lines represent ± 2 standard deviations.

The reviewer suggests that the prior form of the emulator may be important in extrapolation to the regions of the observations. In figure R5 (figure S5 in the supplementary material), we look at the error of prediction for an emulator trained using a constant, or "flat" prior form (our standard emulator is built using a linear model prior). We find that the performance of the emulator is very similar in both situations, suggesting that the prior form is not critical in determining the performance of the emulator in extrapolating at least as far as the observations that we have. We summarise this finding in section 2.1 "The importance of the prior form for emulator predictions" in the supplementary material.

[Figure]

Figure R5/S5. Comparison of emulators for prediction of held-out ensemble members. Black points are the held-out ensemble members, with grey points representing the standard (linear model prior) emulator, and vertical lines ± 2 standard deviations. Orange points represent prediction with a "constant" or "flat" prior, from which the Gaussian process models deviates.

On the reviewer's specific point concerning the results from the one-at-a-time sensitivity analysis (figure 8) being driven by the prior form of the emulator: we find qualitatively similar results if we use a constant prior form for the emulator, suggesting prior form isn't too important. We plot the result in figure R6.

[Figure]

[Figure]

Figure R6. Leave-one-out analysis calculated using a constant or "flat" prior form for the emulator. This is qualitatively similar to that seen in figure 8 of the paper, suggesting that the prior form of the emulator isn't too important. The emulator mea is the solid line, transparent regions represent ± 2 standard deviations.

Reviewer1
Are the results in Fig 9 from the FAST99 algorithm generated by sampling the across the full 2-dimensional climate [P,T] space? How are these results affected by sampling from the emulator where there is no training data, particularly for a cool/wet climate at the top left of Fig 3? Could the sensitivity to the climate variables be over-estimated here?
(I cannot easily tell from the plot, but do the individual main effect sensitivities sum to <1?(Main effect + interaction should sum to 1.) I've seen instances where the algorithm produces main effect values that sum to >1 in the presence of noise in the emulator fit.)

Author response
The reviewer is correct: the FAST99 algorithm samples from the full range of 2 dimensional climate space (temperature and precipitation) and there are no ensemble members in the "cool, wet" corner of that space. The emulator estimate of forest fraction is necessarily an extrapolation, and there is therefore a chance that the sensitivity of rainforest to this corner is

overestimated in the FAST99 sensitivity analysis. As far as we are aware, the Sobol indices calculated by the FAST99 algorithm need to be calculated on the unit hypercube, precluding the option of changing the FAST99 algorithm to take into account the shape of the sample in temperature/precipitation space. We have acknowledged this in the text, and offer another form of sensitivity analysis, in order to place the FAST99 results in context.

First, we address the reviewer's question about the FAST99 indices. The FAST99 algorithm calculates first order effects that sum to 0.982, and the total order effects (including interactions) that sum to 1.114 (see table R1).

| Input | First Order | Total Order |
|---|---|---|
| F0 | 0.034 | 0.046 |
| LAI_MIN | 0.029 | 0.041 |
| NL0 | 0.126 | 0.148 |
| R_GROW | 0.012 | 0.016 |
| TUPP | 0.026 | 0.043 |
| Q10 | 0.000 | 0.003 |
| V_CRIT_ALPHA | 0.314 | 0.347 |
| MOD_TEMP | 0.190 | 0.205 |
| MOD_PRECIP | 0.250 | 0.266 |
| Sum | 0.982 | 1.114 |

Table R1. Sensitivity indices calculated using the FAST99 algorithm.

The reviewer highlights a larger problem of producing an effective sensitivity analysis when there are dependent inputs - for example, in our case when two inputs are not uniformly sampled. At the moment, this is an active research question, with seemingly no perfect standard solution for the circumstances we find ourselves in in this analysis. There appear to be some solutions for the case of dependent or correlated inputs (see e.g. Mara et al., 2015), but these rely on knowing (or assuming a distributional form for) the conditional densities - a situation we are not in, as the conditional density in the T/P space does not appear drawn from a standard distribution. The problem of sensitivity analysis when there is a non-uniform and non hyper-rectangular input space must surely be a priority for the history matching community, as history matching often provides highly irregular constrained input spaces.

We have identified a type of sensitivity analysis that may give useful information on the relative importance of each parameter in determining whether model output is close to

observations. We believe that this may be a good alternative to the variance-based sensitivity of FAST99, and will give modellers useful feedback about the way the system behaves. It also fits neatly into the History Matching framework. The technique is called Monte Carlo Filtering (MCF), or Regional Sensitivity Analysis. A recent description and references can be found in section 3.4 of Pianosi et al. (2016). The basic idea of MCF is to split samples from the input space into those where the corresponding model output meets (or not) some criteria of behaviour. Examining the differences between the cumulative distributions of those inputs where the outputs do or do not meet the criteria provides a measure of sensitivity of the output to that input. For example, we might split model behaviour into those outputs above or below a threshold.

We integrate the MCF sensitivity analysis into the history matching framework. We examine the differences in the univariate distributions of each parameter, in those samples where the output is ruled out by history matching, against those that are "Not Ruled Out Yet" (NROY). To measure the differences between the distributions we perform a two-sided Kolmogorov–Smirnov (KS) test and use the KS statistic as in indicator that the output is sensitive to that input. A larger KS statistic indicates that the cumulative distribution function of the respective inputs are further apart, that that input is more important for determining if the output falls within the NROY part of parameter space, and therefore the output is more sensitive to that input in a critical region. We note that MCF is useful for ranking parameters in order of importance, but not for screening, as inputs that are important only in interactions might have the same NROY and ruled out marginal distributions. In this case they would have a sensitivity index of zero.

We include an MCF analysis in the main text in section 4.4 "Sensitivity analysis", with some supporting analyses in the supplementary material.

We apply MCF in two modes: first, on the 300 model runs in our ensemble, and second, using the emulator fitted to all 300 of those ensemble members. When using the original runs, we remove the problem of sampling from the unit hypercube in temperature and precipitation space. The MCF is calculated using the inputs of the runs as they appear in that space. We simply assign an implausibility measure to each run, and then examine the difference in empirical cumulative distribution functions between those that are ruled out or NROY (implausibility < 3) in each input dimension. We assume zero discrepancy and zero discrepancy uncertainty, and there is zero emulation uncertainty as these are the runs. We assume an observational uncertainty of 0.05 (one standard deviation).

Second, we apply MCF using the emulator. This allows us to examine the difference between distributions given a much larger sample from the input space, and to calculate the uncertainty of the MCF sensitivity indices when calculated using different numbers of runs. This comes at the cost of using an imperfect emulator, which may give different results than if we were using a large ensemble of runs. To avoid the problem of sampling precipitation and temperature from regions where there are no ensemble members, we sample uniformly from across input space for all other parameters, and then append a random temperature/precipitation location from the ensemble.

We calculate a sampling uncertainty by calculating the MCF sensitivity metrics 1000 times, each time using a sample size of between 100 and 3000 points from the input space. In this way, we estimate both the mean and the uncertainty (standard deviation) of that mean, when using a different number of ensemble members to calculate the MCF sensitivity indices, including that for 300 members, our ensemble size. We plot these in figure R7 (also included as supplementary material figure S6).

[Figure]

Figure R7/S6. Mean (left) and standard deviation (right) of the KS statistic the Monte Carlo Filtering index of sensitivity, calculated using different sizes of emulated ensembles.

We note that the sensitivity indices are estimated to be higher when a small number of ensemble members are used, as well as with a higher uncertainty. The change in both the estimated statistic and its uncertainty become small by the time 3000 ensemble members are used, suggesting that we should use at least this many emulated ensemble members to obtain an unbiased sensitivity analysis.

We compare the KS statistics for each input, and for a history matching exercise conducted using each tropical forest observation in figure R8. We plot the KS statistic calculated using only the ensemble members as open points, alongside the estimated uncertainty bounds calculated using 300 emulated members. We plot the KS statistic estimated using 5000 emulated members as solid points and error bars. We include a version of this figure plotting only the estimates using 5000 ensemble members in the main text in figure 15.

[Figure]

Figure R8. Monte Carlo Filtering estimate of sensitivity of model outputs to model inputs. Solid points are estimated using 5000 emulated ensemble members. Open points are estimated using 300 actual ensemble members, with uncertainty estimated using 300 emulated ensemble members.

We can check the strength of the relationship between the MCF sensitivity measures and the FAST99 sensitivity measures, by plotting them together (figure R9, supplementary material S7). We plot only the FAST99 first-order sensitivity, as we do not expect MCF sensitivity to be able to measure interactions between inputs accurately. We find a fairly strong relationship between the two sensitivity measures, although we would expect some differences, as they are measuring different things, and MCF is not sampling from locations in temperature and precipitation space where there are no ensemble members.

The reviewer asked if the FAST99 algorithm might overestimate the sensitivity of forest fraction to temperature and precipitation, due to sampling a corner of input space with no tropical forest. We find some evidence of this being the case, using the emulated MCF sensitivity index, that avoids this issue by only sampling from locations in temperature and precipitation space that exist in the ensemble. The FAST99 algorithm produces very similar sensitivity indices (perhaps fortuitously, as they measure on a different scale) for temperature and precipitation as the MCF algorithm for the Amazon forest, but the Southeast Asian and Central African forests appear less sensitive to these inputs when estimated using the MCF algorithm.

[Figure]

Figure R9/S7. Relationship between the first-order sensitivity of input parameters calculated by the FAST99 algorithm, and that calculated by the Monte Carlo Filtering (MCF) algorithm. The sensitivity indices calculated only using the ensemble members are plotted on the left, with uncertainty estimated by using an emulated 300 member ensemble. On the right, we plot the sensitivity indices when calculated using 5000 emulated ensemble members.

Finally, we can look at the effect of excluding inputs ruled out by history matching has on our leave-one-out sensitivity analysis, with the impact of ruling out some of the higher forest fractions found in the "cool, wet" part of parameter space that is far from any design points. We do this by simply excluding inputs where the calculated implausibility for the outputs is above 3, when assuming observational uncertainty is 0.05 (1 standard deviation), and discrepancy uncertainty is zero. We plot the results in figure R10 (now fig. 14 in the main text, replacing fig. 8 in the original text), and note that very high and very low forest fractions are excluded, somewhat reducing the calculated sensitivity of the forests to various inputs.

[Figure]

[Figure]

Figure R10/14. One-at-a-time sensitivity analysis, excluding any inputs with a calculated implausibility above 3. Semi-transparent regions represent ± 2 standard deviations

A summary of actions taken:
- We moved sensitivity analysis section (now section 4.4) to after the history matching section (now section 4.3).
- We now report only NROY input space in the one-at-a-time analysis (i.e. the history-matched version). Figure 8 in the original text has been replaced with fig. R10, and is now fig. 14 in the updated manuscript.
- We added Monte Carlo Filtering (MCF) results in the main text sensitivity analysis section, and added the comparison of MCF and FAST99 sensitivity indices in the supplementary material.

Reviewer 1
In Figure 10 the emulator is used to simulate across the entire range of simulated temperature and precipitation with all other inputs fixed at the default setting. How might this result be affected?

Author response

We compare our results in figure 10 with an emulator that uses a different form - a constant (or "flat" prior. The different emulator does indeed change the result of the two-at-a-time sensitivity analysis (see figure R11), but not significantly in the areas near design points (i.e. the ensemble members), or in the regions near the forest fraction observations for the three forests. The largest difference is in the "cool, wet" corner of the temperature/precipitation space, well away from ensemble members or observations. We note that figure 10 of the paper contained an error - the background emulated surface was normalised to its own range, not the range that included the original data. We have updated the figure, and it is now moved to figure 8 in the revised main text.

[Figure]

Figure R11. Two-at-a-time sensitivity analysis using the standard emulator with a linear prior form (left), and with a constant or "flat" prior form (right).

Reviewer 1
How might this issue affect the results of the retained parameter space from the history matching (Figures 13-16) for each forest region? In Figure 16, are the regions not covered by the training data more likely to be retained as emulator error is larger, reducing the value of the implausibility metric so that it cannot be ruled out?

Author response
Yes, the issue of extrapolation to unsampled areas of temperature/precipitation parameter space does appear to affect the ruling-out of parameter space. Regions not covered by the training data are more likely to be retained, as the emulator error is larger (but not increasing rapidly - see figure R12). To a large degree, this is the history matching working as it should. There is not enough evidence to rule out inputs where output corresponds to these regions. Recognising that we have other knowledge not accounted for here, this could perhaps be further developed by including an implausibility measure that also includes measures of temperature and precipitation, before bias correction.

We have included the following paragraph at the end of the history matching section (section 4.3):

"It is possible that the estimate of shared NROY input space is larger than it could be, due to the lack of ensemble runs in the ``cool, wet'' part of parameter space, where there are no tropical forests. Inputs sampled from this part of parameter space may not be ruled out, as the uncertainty on the emulator may be large. This is history matching working as it should, as we have not included evidence about what the climate model would do if run in this region. Further work could explore the merits of including information from other sources (for example, from our knowledge that tropical forests do not exist in a cool wet climate) into the history matching process."

[Figure]

Figure R12 Mean (left) and standard deviation of the standard emulator, with all land surface inputs held at their best values, and only temperature and precipitation varied. White regions are "ruled out", as the emulated forest fraction is deemed statistically too distant from any of the observed forests.

Reviewer 1

Page 15 Line 3-4, Figure 15 (and discussion): guiding further runs, choosing high density regions to run new ensemble members. This is a useful outcome. My question here really relates to how the results trace back to improve model performance... How does the bias correction information feed back for the modeller to know what 'atmospheric configuration' should be used (the 10 atmospheric parameters, or beta?) with the inputs selected as good for any new runs? Can a good representation of forest fraction for all forests simultaneously be obtained in new runs at these parameter combinations without bias correction? Or, would any new runs always need to be bias corrected too, until further work to understand the true cause of the climate bias is completed and the climate model updated? As obviously, just running the model at more combinations in this identified joint space will induce climates as shown in Figure 3, away from the observed climates for each forest.Could the authors comment on this in the discussion?

Author response

It would be difficult in this particular case to directly feed back bias correction information to help the modeller select "good" atmospheric and oceanic parameter sets. This is because 1) the original parameter sets for the atmosphere and ocean are not available, 2) only a small selection of the best performing of parameter sets were chosen from a larger set. That larger set was carefully chosen to sample the parameter space effectively, but there is no guarantee that the best-performing sets have this property. In the case that they do not, it would make it difficult to build an accurate emulator of the relationship between the atmospheric and oceanic parameters, the climate bias, and the modelled forest fraction.

It would be simplest to build an emulator that modelled and perhaps minimised directly the climate biases, given the original perturbations of atmospheric and oceanic parameters. The fact that forest fraction seems so strongly influenced by temperature and precipitation suggests that minimising climate biases would be a good way of minimising forest fraction errors. In this paper we have outlined some of the impacts of the climate biases on forest fraction, which should help to motivate efforts to bias correct the climate of the model directly. However, as implied by the results of Gregoire et al. (2010), this might be at the expense of the performance of other parts of the climate model. Informing the modellers of this fact might motivate more work on fundamental structural errors within the model.

Reviewer 1

Further Comments: Page 3 Line 18: 'Without strong prior information...' What is meant by 'prior information' here?What kind of information? On observations? On model skill? On both? This is a bit vague and needs more clarity.

Author response

Added "Without information about known errors (for example, knowledge of an instrument bias, or a known deficiency of a model) …"

Reviewer 1
Page 5 Line 9: What reasons? Please give more details here: '...whereas there were a number of reasons one might reject the proposed parameter space, including...'

Author response
We have restructured the paragraph and given more detail about why we might choose the default parameters over a new region of parameter space.

Reviewer 1
Section 1.3 (Page 5 Line 11): It might give more context to the first listed aim on line 11, and for the detail coming in the second paragraph of the section (discussing the results of McNeall et al (2016), which are not 'aims of this paper') to connect that this study is extending the analysis of McNeall et al. (2016) at the start of the section in the first line (first aim?)?

Author response
We have restructured the section as suggested.

Reviewer 1
Page 6 Line 9-11:Sentence starting 'Parameter perturbations...'. Are there any references for examples of such findings?

Author response
We have added references as follows:

Parameter perturbations and CO2 concentrations have been shown to influence
the simulation of tropical forests in climate models (Boulton et al., 2017; Huntingford et al., 2008), with increases in CO2 fertilisation and associated increased water use efficiency through stomatal closure offsetting the negative impacts of purely climatic changes (Betts et al., 2007; Good et al., 2011).

Reviewer 1
Page 6 Line 19:'...was sensitive to perturbations in parameters,'.This is vague... What kind of parameters? Edit to say '...was sensitive to perturbations in parameters such as...'

Author response
Poulter et al (2010) perturbed 41 parameters they identified as being important in modeling ecosystem processes in LPJmL and affecting the carbon and water cycle, and vegetation dynamics. The criteria they used to estimate the importance of the parameters used a number of model outputs in calculations and so it is difficult to concisely summarise their importance in an overview. We've edited the paragraph to include the fact that the parameters were concerned with ecosystem, carbon, water cycle and vegetation dynamics.

Reviewer 1
Page 6 Line 31 –Page 7 Line 1:I realise that the details of the ensemble are in McNeall et al (2016), but I think it would be useful to give minimal details of the parameters perturbed in this paper also. Their effects are being compared to the climate variables in Section 4.1, with parameter names (acronyms) given in the text, and yet I have to go to a completely different paper to find a description of them /what they correspond to in the model. Please add a small summary table (to the supplementary file, if not to the main paper) that lists the parameters with short descriptions.

Author response
We have added a table of input parameter details to the supplementary material, and added a reference to that in the main body of the text.

Reviewer 1
Page 7 Line 4:What is meant by 'global values'? Global values of what? Please clarify.

Author response
To clarify, we have changed the text to: "The strong relationships between the global mean forest fraction and the mean forest fraction in each region implies that perturbations in input parameters exert a larger control over all forests simultaneously, and individual forests to a smaller extent."

Reviewer 1
Page 8 Line 30: Please provide a reference for GP emulation.

Author response
We have added references to Sacks *et al.* (1989) and Kennedy & O'Hagan (2001).

Page 9 Line 13:'...the 10 atmospheric parameters perturbed in a previous ensemble, summarised by the $\beta$ parameter'. I'm struggling to picture how the effects of 10 parameters can be summarised by 1 parameter. A lot of information is being condensed here? This needs more explanation. (Also see first comment above under 'Specific comments'.)

Author response
We have amended the paragraph to clarify the role of beta, and the atmospheric and oceanic parameters as follows:

"These new inputs are outputs of the model when run at the original inputs X, and are influenced by the 10 atmospheric and oceanic parameters perturbed in a previous ensemble in a configuration unavailable to us in this experiment. Performance of the model under those perturbations is summarised in the parameter, beta which has smaller values for the better performing models. The performance metrics included temperature and precipitation, along with a number of other measures so the beta parameter therefore contains information about temperature and precipitation across the ensemble, without being a perfect representation of its behaviour. We cannot control the atmospheric and oceanic parameters directly and thus ensure that they lie in a latin hypercube configuration, although the

ensemble is ordered in a latin hypercube configuration according to the performance of the model at each parameter set."

Reviewer 1
Page 9 Line 14: This sentence: 'We cannot control them directly and thus ensure that they lie in a latin hypercube configuration' is confusing and could be interpreted in different ways. I first read 'and thus ensure' as that you do ensure that they lie in a LH configuration. But on second read I see the meaning you want is that you can't ensure this. Please re-phrase.

Author response
Rephrased (see above).

Reviewer 1
Also, the 'latin' in 'Latin hypercube' should have a capitol L. Please update here and elsewhere.

Author response
Corrected.

Reviewer 1
Page 10 Line 12: '3% of the maximum possible value of the ensemble.' What does this really statement tell us? The maximum forest fraction is 1 so the error of 0.03 is 3% of this forest fraction value, but this is the minimum percentage of an output value that it could be. The majority of predictions will be less than 1, and so the error of an 'average prediction' is in general a larger percentage than this. It seems a bit of a misleading statement, and I suggest removing it here and on line 14.

Author response
The inclusion of the 3% figure was an attempt to offer some context and concreteness for a small number (0.03). We take the reviewer's point that this choice might flatter the emulator somewhat, and therefore present the number as a proportion of the mean forest fraction (0.54), meaning the figure is around 6%, for the augmented emulator and 12% for the standard emulator.

Reviewer 1
Page 10 Line 25: I don't understand how the 'rank histogram' indicates that we have 'reliable' uncertainty estimates? It shows the predictions are a mixture of over and under-estimations of the actual model, but it gives no indication of the size of errors, which could be large and therefore not reliable? Maybe I misunderstand this.

Author response
Although the reviewer is correct that the rank histogram does not address the size of the errors, it does ensure that the size of the errors are consistent with the predicted error distributions. Therefore, the size of the errors is not beyond what we thought it should be (this is what we mean by "reliable"). This information, coupled with the fact that the errors themselves are not large should be enough to have confidence both in the accuracy of the

emulator, and its uncertainty estimates. We have added the sentence "Do the error estimates of the augmented emulator match the true error distributions when tested in leave-one-out predictions?" at the beginning of this section to make the intention of the analysis clearer.

Reviewer 1
Section 4.2 and Figure 10: The interpretation of this plot needs clarification. Would the contours be similar at different parts of the land-surface parameter space? (Fixing at a different simulation to the default?) On Page 11, Line 26, it suggests that for central Africa, moving any ensemble member (small triangle point) to the observed (big triangle) would not cross many contours, but the central Africa points with wetter climates (normalised T at approx. 0.4, normalised P at approx. 0.5 to 0.6) would cross between 3 and 4 contours to be in the same one as the observed (big triangle), so I'm not sure this is true? Please clarify this.

Author response
On the first point, the reviewer is correct, there would be some differences if the land surface parameters were fixed at a different point. However, 1) the analysis of this paper supports the conclusion that the default parameters for the land surface are perfectly valid, and 2) given that they are unlikely to be large (sensitivity analysis indicates small interaction terms) we question whether summarising a very large number of possible perturbations will add value to the analysis. On the second point, the reviewer is correct. We have amended the paragraph as below, to make clear that we are talking about a "typical" ensemble member (i.e. one from the centre of each forest sub-ensemble), rather than about any of the ensemble members.

Updated paragraph:
"Moving an indicative ensemble member from the centre of these forest sub-ensembles to observed values of temperature and precipitation would shift them primarily in the same direction as the contours of forest fraction value. South East Asian and Central African ensemble members are  therefore simulated with a roughly accurate forest fraction. In contrast, the Amazon is simulated slightly drier, and considerably warmer than the observed Amazon and many ensemble members consequently have a lower forest fraction than observed. Shifting a typical ensemble member for the Amazon to its observed temperature and precipitation would cross a number of contours of forest fraction.  This figure provides strong evidence that a significant fraction of the bias in Amazon forest fraction is caused by a bias in simulated climate."

Reviewer 1
Page 12 Line 8-11: Are the values given in this paragraph mean absolute error between model and observations? The first line says 'difference', but this must be an average? Please clarify. Also, could the lack of training data near the observed climate for central Africa be a contributing factor to why central Africa is worse (Line 9) in this metric?

Author response

The numbers given in this paragraph are the difference between observations and 1) predictions of forest fraction at the default land surface parameters using a standard emulator and 2) predictions of the forest fraction at the default land surface parameters and the corrected local climate, using the augmented emulator. We have restructured the paragraph to make it clearer exactly what we have done. The reviewer makes a good point about the lack of training data near the central African forest, and we have included this. The paragraph now reads:

"The bias correction reduces the difference between the prediction for the modelled and observed Amazon forest fraction markedly, from -0.28 using the standard emulator to -0.08 using the augmented emulator. It makes the predicted modelled forest in central Africa worse (-0.11 from -0.03), and slightly improves the SE Asian forest fraction (0.07 from 0.1). Overall, bias correcting the climate takes the mean absolute error at the default parameters from 0.14 to 0.09 for the three forests. It is possible that the predicted forest fraction for central Africa is slightly worse because the observed climate is towards the edge of the parameter space of temperature and precipitation, and there are no runs near."

Reviewer 1
Section 4.4: It might be better for the flow of the results section if the first part of Section 4.4 (up to Page 13 Line 2) describing the history matching methodology was moved into Section 3 on methods?

Author response
Amended as requested.

Reviewer 1
Page 13 Line 12:Could more detail be given as to why the implausibility value at the default settings rises for central Africa and SE Asia on bias correction?

Author response
This is due to the slight rise in predicted error in the case of central Africa, and in the reduction of uncertainty in the case of SE Asia. Both of these would be expected to raise the Implausibility score somewhat. The paragraph has been amended to acknowledge this.

Reviewer 1
Page 17 Line 14-16: Summarising the 10 parameters using 2 outputs is useful, but this is likely to have little traceability back to the original 10 input values? Many different combinations of the 10 inputs could lead to a similar combination in the 2 outputs, so how would one know what combination of the 10 inputs is best when setting up any further runs of the model? Also, the 'O(10xp)' rule is when training points are space-filling across the parameter space being emulated. There is no guarantee (as seen in this example) that this property will hold when outputs are used as dimension reduction, so more points, or even less points, could easily be required. This should be acknowledged.

Author response

The reviewer is correct that there is likely little traceability back to the original 10 input values, and the original ensemble must be examined to get that information. To address the reviewer's points we have added the following paragraph:

"We acknowledge however that in order to trace back information about the performance of the model in forest fraction to the original 10 oceanic and atmospheric parameters, we would need access to the original ensemble. We have used temperature and precipitation to reduce the dimension of the parameter space, but there is no guarantee that the relationship between the original parameters and the local climate is unique. There may be multiple combinations of the 10 parameters that lead to the temperature and precipitation values seen, which would mean that we would require a large ensemble to estimate the relationships well. Alternatively, there may be an even more efficient dimension reduction for forest fraction, meaning we would need even fewer model runs to summarise the relationship."

**Technical corrections:**

Reviewer 1
Page 1 Line 13-14: 'This might be due to...'. I think this sentence might be easier to read if the number/list format is replaced with 'This might be due to either ..., or...., or a combination of both.'

Response: Amended as requested.

Reviewer 1
Page 1 Line 16-17: '...alongside regular land surface input parameters.'.Is the term 'regular' needed here? Maybe remove this word. [The 'regular' suggests to me that there may be other types of land surface input parameters that are 'not regular' which are not included, which I don't think is the case.]

Response: Amended as requested.

Reviewer 1
Page 1 line 15: Should '...a climate model...' be '...the climate model...'?This is now the specific climate model used by McNeall et al (2016).

Response: Amended to clarify that we are using the augmented emulator on the specific climate model from McNeall (2016).

Reviewer 1
Page 1 Line 17-18 :For readability, please change 'is nearly as sensitive to climate variables as changes in its land surface parameter values.' to 'is nearly as sensitive to climate variables as it is to changes in land surface parameter values.'

Response: Amended as requested.

Reviewer 1
Page 2 Line 9: Should '...processes sufficiently to trust...' be '...processes sufficiently and to trust...'?

Response:  Added "well", so the sentence now reads  "We wish to choose input parameters where the output of the model reproduces observations of the climate, in order to have confidence that the model represents important physical processes sufficiently well to trust projections of the future."

Reviewer 1
Page 2 Line 28:The sentence here is hard to read. Change: '...practices, there appear no standard procedures for climate model tuning however -as the authors...' to'...practices, there appear to be no standard procedures for climate model tuning. However, as the authors...'.

Response: The section is amended for clarity.

Reviewer 1
Page 2 Line 32: Missing word? Edit: 'It might start with single column version...' to be 'It might start with a single column version...'

Response: Amended as requested.

Reviewer 1
Page 3 Line 2: Change word order? Edit: '...might be then tuned...' to '...might then be tuned...'

Response: Amended as requested.

Reviewer 1
Page 3 Line 8: Change 'Golaz et al. (2013) Show...' to 'Golaz et al. (2013) show...'.

Response: Amended as requested.

Reviewer 1
Page 3 Line 20-21: This sentence needs plural 'candidates' at the start, and the second part should be given as more of a negative to make the point? Revise to: 'This means that good candidates for input parameters might be found in a large volume of input space, but projections of the model made with candidates from across that space might diverge to display a very wide range of outcomes.'

Response: Amended as requested.

Reviewer 1
Page 3 Line 23: 'individual parts' Of the tuning process? Or the model? Please clarify.

Response: Clarified that this meant individual parts of the process.

Reviewer 1
Page 4 line 14: Missing full stop after Vernon et al. (2010).

Response: Corrected.

Reviewer 1
Page 4 Line 20, Line 23: References in bracketed format when should be in in-line format.

Response: Corrected.

Reviewer 1
Page 4 line 33: Remove the second 'used'.

Response: Corrected.

Reviewer 1
Page 5 line 1: Missing words. Change to: '...structural bias in the ocean component of the climate model HadCM3 could'

Response: Corrected.

Reviewer 1
Page 5 Line 33: Should this be '...use the augmented emulator to estimate the sensitivity...'?

Response: Yes it should, corrected.

Reviewer 1
Page 7 Line 2: Move the reference to Fig 1 to the next sentence, which is the sentence that is describing what is shown in Fig 1.

Response: Corrected.

Reviewer 1
Page 7 Line 7-8: '...parameter settings which the emulator suggested should lead to an adequate simulations of...'. The emulator provides the predictions of model output but does not indicate adequacy –this comes from the history matching process (as the authors have described). Change to: '...parameter settings which the history matching process suggested should lead to adequate simulations of...'.

Response: Amended as requested.

Reviewer 1
Page 8 Line 9, 12: References to Jones et al, and Adler et al are in in-line format when should be in bracketed format?

Response: Amended as requested.

Reviewer 1
Page 9 Line 10: Change to:'...each of the forests:the Amazon, central Africa and Southeast Asia...' or put the forest region names in brackets?

Response: Added a colon.

Reviewer 1
Page 9 Line 11-12: For readability, move the sentence 'Regional extent of ... supplementary material.' so that it is the second sentence in this paragraph. (The next sentence follows better from the one before it!)

Response: Moved as requested.

Reviewer 1
Page 9 Line 24: Here the work by M16 is referred to as being by the authors of this work ('we built'), but in all previous references to this point it has been referred to as a separate study (e.g. M16 argue..., or M16 speculated...). Update as needed to be consistent.

Response: Amended as requested.

Reviewer 1
Page 11 Line 3: Should this paragraph start with: 'The augmented emulator...'

Response: Amended as requested.

Reviewer 1
Page 11 Line 5: '...predict changes in forest fraction as each variable is changed from the lowest to highest setting in turn...'. Should 'variable' in this sentence be replaced with 'input'? – as this is done for the land surface input parameters as well as the climate variable inputs?

Response: Changed as requested, and updated to refer to the augmented emulator.

Reviewer 1
Page 12 Line 22: Remove the second 'the'.

Response: Corrected.

Reviewer 1
Page 13 Line 21:The term 'the climate-bias forest' sounds weird? Should this be 'the climate-bias-corrected forest'?

Response: Yes, Corrected.

Reviewer 1
Page 16 Line 34: Remove the second 'could'.

Response: Corrected.

Reviewer 1
Page 17 Line 14: 'O(170)' should be in italics?

Response: Yes, Corrected.

Reviewer 1
Page 18 Line 6: Change: 'If trained an ensemble...' to 'If trained on an ensemble...'. Also, remove the second 'which'.

Response: Corrected.

Reviewer 1
Page 18 Line 10: Remove; '(e.g.)'.

Response: Amended as requested.

Reviewer 1
Page 18 Line 19: Should 'learned' be 'learn'?

Response: Yes, Corrected.

Reviewer 1
Page 18 Line 32: Missing word. Change: '...in leave-one-out...' to '...in a leave-one-out...'.

Response: Corrected.

Reviewer 1
Page 19 Line 13:Change 'finding' to 'findings'.

Response: Corrected.

Reviewer 1
Page 27, Fig 5 caption: The brackets for g1are in the wrong place? For gnand yn, should 'n' be replaced with '3', as is shown in the diagram, and written for 'C' in the next sentence?

Response: Yes, both corrected.

Reviewer 1
Supplementary information Line 17: Reference in bracketed format when should be in in-line format.

Response: Corrected.

Reviewer 1
Supplementary information Line 18: Missing word. Change: '...in section 3 the...' to '...in section 3 of the...'

Response: Corrected.

**Reviewer 2**

The authors seek to produce an emulator that can account for some of the known biases in a climate model when the aim is to find a 'good' parameter set to represent observations. Overall, I think the idea is a good one and the augmenting of the emulator in this case clearly works to make a better emulator for model constraint. The paper is well written and the method easy to follow. The work should be published in GMD. I have a few points to discuss:

I am pretty confused about the 'beta' parameter and what it means in both the original model and the emulator here – can this be clarified in the text please.

Author Response
Reviewer 1 had a very similar comment, we have comprehensively responded in the first response to reviewer 1, adding expanded explanation of the beta parameter at the end of section 2.1 "Biases in FAMOUS".

Reviewer 2
How much do you need to know about the present bias? It's clear the authors had information on this and a good idea from the modellers where the biases came from but it's less clear what they may have done with less information.

Author response
Knowledge about the present bias in (e.g.) temperature and precipitation is certainly useful when it comes to understanding where the model may be wrong, and in attributing the bias in forest fraction to climate rather than model inadequacy. To gain a good insight into what is possible with similar tools, but less understanding of the sources of bias in the land surface model, we refer the reviewer to the analysis of McNeall *et al.* (2016), which was conducted without this understanding. In particular, the understanding of the shape of the combined parameter space that could not be ruled out, and that the default land surface parameters are within this region are knowledge gained from an understanding of the impacts of the climate bias in this paper.

Reviewer 2
Page 11, Line 26: 'Moving any...' I find this sentence confusing. Can you better link it to the figure and clarify?

Author response

We have edited the paragraph for clarity, and it now reads as follows -

"With other inputs held constant, cooler, wetter climates are predicted to increase forest fraction and drier, warmer climates reduce forest fraction. In general, South East Asian and Central African forests are simulated as warmer and wetter than their true-life counterparts. Moving the temperature and precipitation values of a typical ensemble member from near the centre of these forest sub-ensembles to their observed real-world values would shift them primarily in the same direction as the contours of forest fraction value. This would mean that bias correcting the climate variables would not have a large impact on forest fraction values in South East Asian and Central African forests, and that they are therefore simulated with a roughly accurate forest fraction. In contrast, the Amazon is simulated slightly drier, and considerably warmer than the observed Amazon and many ensemble members consequently have a lower forest fraction than observed. Shifting the temperature and precipitation of a typical ensemble member for the Amazon to its real-world observed values would cross a number of contours of forest fraction.  This figure provides strong evidence that a significant fraction of the bias in Amazon forest fraction is caused by a bias in simulated climate."

Reviewer 2
Figure 14: There is a clear relationship with V_CRIT_ALPHA and NLO when the augmented emulator is used. Can you discuss this and what it might mean?

Response: It is clear from the sensitivity analyses section that, all other things remaining the same, (e.g.) increasing the value of NL0 raises strongly forest fraction, while increasing V_CRIT_ALPHA strongly reduces forest fraction. We would expect there to be a region, indeed a plane through parameter space where these two strong effects counteract each other, resulting in a forest fraction close to observations.

This feature does not appear in the history matching before bias correcting (figure 13 in the original text). The low value of the simulated Amazon forest fraction before bias correction of the climate inputs rules out much of the input parameter space later found to be Not Ruled Out Yet (NROY) after the bias corrected history matching exercise (figure 14 in the original text). This point has been added to the discussion.

Reviewer 2
Page 17, line 5: Is it temperature and precipitation that should be targeted or how the model treats them? It's not clear to me exactly what you are recommending.

Author response
Edited for clarity. The sentence now reads "Our work here shows this process as an example. We have identified the importance of precipitation and temperature to the correct simulation of the Amazon forest, and flag their accurate simulation in that region as a priority in for the development of any climate model that hopes to simulate the forest well."

Reviewer 2

Page 17, line 18: If you were to this for every grid box would you expect predictability?I could see this method working for elevation and seasonal biases? How might you go about this?

Author response
We have edited the paragraph to further discuss the practicality and predictability of using the emulator at smaller scales, as follows:

"In theory the augmented emulator could be used to bias correct differently sized regions, down to the size of an individual gridbox for a particular variable. This might be useful for correcting, for example, known biases in elevation or seasonal climate. The principle of repeating the common parameter settings in the design matrix, and including model outputs as inputs would work in exactly the same way, but with a larger number of repeated rows.

In the case of using an augmented emulator on a per-gridbox basis, we might expect the relationship between inputs that we are bias correcting (e.g. temperature, precipitation), and the output of interest (e.g. forest fraction) to be a less clear, as at small scales there are potentially many other inputs that might influence the output. An emulator for an individual gridbox might therefore be less accurate. However, with enough data points, or examples (and there would be many), we might expect to be able to recover any important relationships."

**References not in main text**

Mara, T.A., Tarantola, S. and Annoni, P., 2015. Non-parametric methods for global sensitivity analysis of model output with dependent inputs. *Environmental modelling & software*, *72*, pp.173-183. https://doi.org/10.1016/j.envsoft.2015.07.010